# Local SGD with Periodic Averaging: Tighter Analysis and Adaptive Synchronization

**Farzin Haddadpour**
Penn State
fxh18@psu.edu

**Mohammad Mahdi Kamani**
Penn State
mqk5591@psu.edu

**Mehrdad Mahdavi**
Penn State
mzm616@psu.edu

**Viveck R. Cadambe**
Penn State
vxc12@psu.edu

## Abstract

Communication overhead is one of the key challenges that hinders the scalability of distributed optimization algorithms. In this paper, we study local distributed SGD, where data is partitioned among computation nodes, and the computation nodes perform local updates with periodically exchanging the model among the workers to perform averaging. While local SGD is empirically shown to provide promising results, a theoretical understanding of its performance remains open. We strengthen convergence analysis for local SGD, and show that local SGD can be far less expensive and applied far more generally than current theory suggests. Specifically, we show that for loss functions that satisfy the Polyak-Łojasiewicz condition, $O((pT)^{1/3})$ rounds of communication suffice to achieve a linear speed up, that is, an error of $O(1/pT)$, where $T$ is the total number of model updates at each worker. This is in contrast with previous work which required higher number of communication rounds, as well as was limited to strongly convex loss functions, for a similar asymptotic performance. We also develop an adaptive synchronization scheme that provides a general condition for linear speed up. Finally, we validate the theory with experimental results, running over AWS EC2 clouds and an internal GPU cluster.

## 1 Introduction

We consider the problem of distributed empirical risk minimization, where a set of $p$ machines, each with access to a different local shard of training examples $\mathcal{D}_i, i = 1, 2, , \ldots, p$, attempt to jointly solve the following optimization problem over entire data set $\mathcal{D} = \mathcal{D}_1 \cup \ldots \cup \mathcal{D}_p$ in parallel:

$$\min_{\mathbf{x} \in \mathbb{R}^d} F(\mathbf{x}) \triangleq \frac{1}{p} \sum_{i=1}^{p} f(\mathbf{x}; \mathcal{D}_i), \tag{1}$$

where $f(\cdot; \mathcal{D}_i)$ is the training loss over the data shard $\mathcal{D}_i$. The predominant optimization methodology to solve the above optimization problem is stochastic gradient descent (SGD), where the model parameters are iteratively updated by

$$\mathbf{x}^{(t+1)} = \mathbf{x}^{(t)} - \eta \tilde{\mathbf{g}}^{(t)}, \tag{2}$$

where $\mathbf{x}^{(t)}$ and $\mathbf{x}^{(t+1)}$ are solutions at the $t$th and $(t + 1)$th iterations, respectively, and $\tilde{\mathbf{g}}^{(t)}$ is a stochastic gradient of the cost function evaluated on a small mini-batch of all data.

In this paper, we are particularly interested in synchronous distributed stochastic gradient descent algorithms for non-convex optimization problems mainly due to their recent successes and popularity in deep learning models [26, 29, 46, 47]. Parallelizing the updating rule in Eq. (2) can be

Table 1: Comparison of different local-SGD with periodic averaging based algorithms.

| Strategy | Convergence Rate | Communication Rounds $(T/\tau)$ | Extra Assumption | Setting |
|---|---|---|---|---|
| [43] | $O\left(\frac{G^2}{\sqrt{pT}}\right)$ | $O\left(p^{\frac{3}{4}}T^{\frac{3}{4}}\right)$ | Bounded Gradients | Non-convex |
| [38] | $O\left(\frac{1}{\sqrt{pT}}\right)$ | $O\left(p^{\frac{3}{2}}T^{\frac{1}{2}}\right)$ | No | Non-convex |
| [33] | $O\left(\frac{G^2}{pT}\right)$ | $O\left(p^{\frac{1}{2}}T^{\frac{1}{2}}\right)$ | Bounded Gradients | Strongly Convex |
| **This Paper** | $O\left(\frac{1}{pT}\right)$ | $O\left(p^{\frac{1}{3}}T^{\frac{1}{3}}\right)$ | **No** | **Non-convex under PL Condition** |

done simply by replacing $\tilde{\mathbf{g}}^{(t)}$ with the average of partial gradients of each worker over a random mini-batch sample of its own data shard. In fully synchronous SGD, the computation nodes, after evaluation of gradients over the sampled mini-batch, exchange their updates in every iteration to ensure that all nodes have the same updated model. Despite its ease of implementation, updating the model in fully synchronous SGD incurs significant amount of communication in terms of *number of rounds* and *amount of data exchanged* per communication round. The communication cost is, in fact, among the primary obstacles towards scaling distributed SGD to large scale deep learning applications [30, 4, 44, 21]. A central idea that has emerged recently to reduce the communication overhead of vanilla distributed SGD, while preserving the linear speedup, is *local SGD*, which is the focus of our work. In local SGD, the idea is to perform *local* updates with periodic averaging, wherein machines update their own local models, and the models of the different nodes are averaged periodically [43, 38, 51, 23, 45, 49, 33]. Because of local updates, the model averaging approach reduces the number of communication rounds in training and can, therefore, be much faster in practice. However, as the model for every iteration is not updated based on the entire data, it suffers from a residual error with respect to fully synchronous SGD; but it can be shown that if the averaging period is chosen properly the residual error can be compensated. For instance, in [33] it has been shown that for strongly convex loss functions, with a fixed mini-batch size with local updates and periodic averaging, when $T$ model update iterations are performed at each node, the linear speedup of the parallel SGD is attainable only with $O\left(\sqrt{pT}\right)$ rounds of communication, with each node performing $\tau = O(\sqrt{T/p})$ local updates for every round. If $p < T$, this is a significant improvement than the naive parallel SGD which requires $T$ rounds of communication. This motivates us to study the following key question: *Can we reduce the number of communication rounds even more, and yet achieve linear speedup?*

In this paper, we give an affirmative answer to this question by providing a tighter analysis of local SGD via model averaging [43, 38, 51, 23, 45, 49]. By focusing on possibly non-convex loss functions that satisfy smoothness and the Polyak-Łojasiewicz condition [18], and performing a careful convergence analysis, we demonstrate that $O((pT)^{1/3})$ rounds of communication suffice to achieve linear speed up for local SGD. To the best of our knowledge, this is the first work that presents bounds better than $O\left(\sqrt{pT}\right)$ on the communication complexity of local SGD with fixed minibatch sizes - our results are summarized in Table 1.

The convergence analysis of periodic averaging, where the models are averaged across nodes after every $\tau$ local updates was shown at [49], but it did not prove a linear speed up. For non-convex optimization [43] shows that by choosing the number of local updates $\tau = O\left(T^{\frac{1}{4}}/p^{\frac{3}{4}}\right)$, model averaging achieves linear speedup. As a further improvement, [38] shows that even by removing bounded gradient assumption and the choice of $\tau = O\left(T^{\frac{1}{2}}/p^{\frac{3}{2}}\right)$, linear speedup can be achieved for non-convex optimization. [33] shows that by setting $\tau = O\left(T^{\frac{1}{2}}/p^{\frac{1}{2}}\right)$ linear speedup can be achieved by $O\left(\sqrt{pT}\right)$ rounds of communication. The present work can be considered as a tightening of the aforementioned known results. In summary, the main contributions of this paper are highlighted as follows:

- We improve the upper bound over the number of local updates in [33] by establishing a linear speedup $O\left(1/pT\right)$ for **non-convex optimization problems** under **Polyak-Łojasiewicz** condition with $\tau = O\left(T^{\frac{2}{3}}/p^{\frac{1}{3}}\right)$. Therefore, we show that $O\left(p^{\frac{1}{3}}T^{\frac{1}{3}}\right)$ communication rounds are sufficient, in contrast to previous work that showed a sufficiency of $O\left(\sqrt{pT}\right)$. Importantly, our analysis does not require boundedness assumption for stochastic gradients unlike [33].

- We introduce an adaptive scheme for choosing the communication frequency and elaborate on conditions that linear speedup can be achieved. We also empirically verify that the adaptive scheme outperforms fix periodic averaging scheme.
- Finally, we complement our theoretical results with experimental results on Amazon EC2 cluster and an internal GPU cluster.

## 2 Other Related Work

**Asynchronous parallel SGD.** For large scale machine learning optimization problems parallel mini-batch SGD suffers from synchronization delay due to a few slow machines, slowing down entire computation. To mitigate synchronization delay, *asynchronous* SGD method are studied in [28, 8, 19]. These methods, though faster than synchronized methods, lead to convergence error issues due to stale gradients. [2] shows that limited amount of delay can be tolerated while preserving linear speedup for convex optimization problems. Furthermore, [50] indicates that even polynomially growing delays can be tolerated by utilizing a quasilinear step-size sequence, but without achieving linear speedup.

**Gradient compression based schemes.** A popular approach to reduce the communication cost is to decrease the number of transmitted bits at each iteration via gradient compression. Limiting the number of bits in the floating point representation is studied at [8, 13, 25]. In [4, 40, 44], random quantization schemes are studied. Gradient vector sparsification is another approach analyzed in [4, 40, 39, 5, 30, 34, 9, 3, 36, 21, 32].

**Periodic model averaging.** The *one shot* averaging, which can be seen as an extreme case of model averaging, was introduced in [51, 23]. In these works, it is shown empirically that one-shot averaging works well in a number of optimization problems. However, it is still an open problem whether the one-shot averaging can achieve the linear speed-up with respect to the number of workers. In fact, [45] shows that one-shot averaging can yield inaccurate solutions for certain non-convex optimization problems. As a potential solution, [45] suggests that more frequent averaging in the beginning can improve the performance. [48, 31, 11, 16] represent statistical convergence analysis with only one-pass over the training data which usually is not enough for the training error to converge. Advantages of model averaging have been studied from an empirical point of view in [27, 7, 24, 35, 17, 20]. Specifically, they show that model averaging performs well empirically in terms of reducing communication cost for a given accuracy. Furthermore, for the case of $T = \tau$ the work [16] provides speedup with respect to bias and variance for the quadratic square optimization problems. There is another line of research which aims to reduce communication cost by adding data redundancy. For instance, reference [15] shows that by adding a controlled amount of redundancy through coding theoretic means, linear regression can be solved through one round of communication. Additionally, [14] shows an interesting trade-off between the amount of data redundancy and the accuracy of local SGD for general non-convex optimization.

**Parallel SGD with varying minbatch sizes.** References [10, 6] show, for strongly convex stochastic minimization, that SGD with exponentially increasing batch sizes can achieve linear convergence rate on a single machine. Recently, [42] has shown that remarkably, with exponentially growing mini-batch size it is possible to achieve linear speed up (i.e., error of $O(1/pT)$) with only $\log T$ iterations of the algorithm, and thereby, when implemented in a distributed setting, this corresponds to $\log T$ rounds of communication. The result of [42] implies that SGD with exponentially increasing batch sizes has a similar convergence behavior as the full-fledged (non-stochastic) gradient descent. While the algorithm of [42] provides a different way of reducing communication in distributed setting, for a large number of iterations, their algorithm will require large mini-

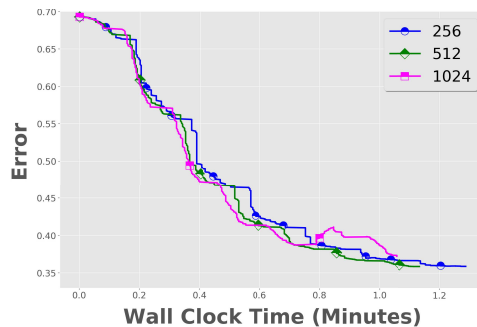

Figure 1: Running SyncSGD for different number of mini-batches on Epsilon dataset with logistic regression. Increasing mini-batches can result in divergence as it is the case here for mini-batch size of 1024 comparing to mini-batch size of 512. For experiment setup please refer to Section 6. A similar observation can be found in [20].

---

**Algorithm 1** LUPA-SGD($\tau$): Local updates with periodic averaging.

1:     **Inputs:** $\mathbf{x}^{(0)}$ as an initial global model and $\tau$ as averaging period.
2:     **for** $t = 1, 2, \ldots, T$ **do**
3:         **parallel for** $j = 1, 2, \ldots, p$ **do**
4:         $j$-th machine uniformly and independently samples a mini-batch $\xi_j^{(t)} \subset \mathcal{D}$ at iteration $t$.
5:         Evaluates stochastic gradient over a mini-batch, $\tilde{\mathbf{g}}_j^{(t)}$ as in (3)
6:         **if** $t$ divides $\tau$ **do**
7:           $\mathbf{x}_j^{(t+1)} = \frac{1}{p} \sum_{j=1}^{p} \left[ \mathbf{x}_j^{(t)} - \eta_t\, \tilde{\mathbf{g}}_j^{(t)} \right]$
8:         **else do**
9:           $\mathbf{x}_j^{(t+1)} = \mathbf{x}_j^{(t)} - \eta_t\, \tilde{\mathbf{g}}_j^{(t)}$
10:         **end if**
11:         **end parallel for**
12: **end**
13: **Output:** $\bar{\mathbf{x}}^{(T)} = \frac{1}{p} \sum_{j=1}^{p} \mathbf{x}_j^{(T)}$

---

batches, and washes away the computational benefits of the *stochastic* gradient descent algorithm over its deterministic counter part. Furthermore certain real-world data sets, it is well known that larger minibatches also lead to poor generalization and gradient saturation that lead to significant performance gaps between the ideal and practical speed up [12, 22, 41, 20]. Our own experiments also reveal this (See Fig. 1 that illustrates this for a logistic regression and a fixed learning rate). Our work is complementary to the approach of [42], as we focus on approaches that use local updates with a fixed minibatch size, which in our experiments, is a hyperparameter that is tuned to the data set.

# 3   Local SGD with Periodic Averaging

In this section, we introduce the local SGD with model averaging algorithm and state the main assumptions we make to derive the convergence rates.

**SGD with Local Updates and Periodic Averaging.** Consider a setting with training data as $\mathcal{D}$, loss functions $f_i : \mathbb{R}^d \to \mathbb{R}$ for each data point indexed as $i \in 1, 2, \ldots, |\mathcal{D}|$, and $p$ distributed *machines* . Without loss of generality, it will be notationally convenient to assume $\mathcal{D} = \{1, 2, \ldots, |\mathcal{D}|\}$ in the sequel. For any subset $\mathcal{S} \subseteq \mathcal{D}$, we denote $f(\mathbf{x}, \mathcal{S}) = \sum_{i \in \mathcal{S}} f_i(\mathbf{x})$ and $F(\mathbf{x}) = \frac{1}{p} f(\mathbf{x}, \mathcal{D})$. Let $\xi$ denote a $2^{|\mathcal{D}|} \times 1$ random vector that encodes a subset of $\mathcal{D}$ of cardinality $B$, or equivalently, $\xi$ is a random vector of Hamming weight $B$. In our *local updates with periodic averaging* SGD algorithm, denoted by LUPA-SGD($\tau$) where $\tau$ represents the number of local updates, at iteration $t$ the $j$th machine samples mini-batches $\xi_j^{(t)}$, where $\xi_j^{(t)}, j = 1, 2, \ldots, p, t = 1, 2, \ldots, \tau$ are independent realizations of $\xi$. The samples are then used to calculate stochastic gradient as follows:

$$\tilde{\mathbf{g}}_j^{(t)} \triangleq \frac{1}{B} \nabla f(\mathbf{x}_j^{(t)}, \xi_j^{(t)}) \tag{3}$$

Next, each machine, updates its own local version of the model $\mathbf{x}_j^{(t)}$ using:

$$\mathbf{x}_j^{(t+1)} = \mathbf{x}_j^{(t)} - \eta_t\, \tilde{\mathbf{g}}_j^{(t)} \tag{4}$$

After every $\tau$ iterations, we do the model averaging, where we average local versions of the model in all $p$ machines. The pseudocode of the algorithm is shown in Algorithm 1. The algorithm proceeds for $T$ iterations alternating between $\tau$ local updates followed by a communication round where the local solutions of all $p$ machines are aggregated to update the global parameters. We note that unlike parallel SGD that the machines are always in sync through frequent communication, in local SGD the local solutions are aggregated every $\tau$ iterations.

**Assumptions.** Our convergence analysis is based on the following standard assumptions. We use the notations $\mathbf{g}(\mathbf{x}) \triangleq \nabla F(\mathbf{x}, \mathcal{D})$ and $\tilde{\mathbf{g}}(\mathbf{x}) \triangleq \frac{1}{B} \nabla f(\mathbf{x}, \xi)$ below. We drop the dependence of these functions on $\mathbf{x}$ when it is clear from context.

**Assumption 1** (Unbiased estimation). *The stochastic gradient evaluated on a mini-batch $\xi \subset \mathcal{D}$ and at any point $\mathbf{x}$ is an unbiased estimator of the partial full gradient, i.e. $\mathbb{E}\left[\tilde{\mathbf{g}}(\mathbf{x})\right] = \mathbf{g}(\mathbf{x})$ for all $\mathbf{x}$.*

**Assumption 2** (Bounded variance [6]). *The variance of stochastic gradients evaluated on a mini-batch of size $B$ from $\mathcal{D}$ is bounded as*

$$\mathbb{E}\left[\|\tilde{\mathbf{g}} - \mathbf{g}\|^2\right] \leq C_1\|\mathbf{g}\|^2 + \frac{\sigma^2}{B} \tag{5}$$

*where $C_1$ and $\sigma$ are non-negative constants.*

Note that the bounded variance assumption (see [6]) is a stronger form of the above with $C_1 = 0$.

**Assumption 3** ($L$-smoothness, $\mu$-Polyak-Łojasiewicz (PL)). *The objective function $F(\mathbf{x})$ is differentiable and $L$-smooth: $\|\nabla F(\mathbf{x}) - \nabla F(\mathbf{y})\| \leq L\|\mathbf{x} - \mathbf{y}\|, \forall \mathbf{x}, \mathbf{y} \in \mathbb{R}^d$, and it satisfies the Polyak-Łojasiewicz condition with constant $\mu$: $\frac{1}{2}\|\nabla F(\mathbf{x})\|_2^2 \geq \mu\left(F(\mathbf{x}) - F(\mathbf{x}^*)\right), \forall \mathbf{x} \in \mathbb{R}^d$ with $\mathbf{x}^*$ is an optimal solution, that is, $F(\mathbf{x}) \geq F(\mathbf{x}^*), \forall \mathbf{x}$.*

**Remark 1.** *Note that the PL condition does not require convexity. For instance, simple functions such as $f(x) = \frac{1}{4}x^2 + \sin^2(2x)$ are not convex, but are $\mu$-PL. The PL condition is a generalization of strong convexity, and the property of $\mu$-strong convexity implies $\mu$-Polyak-Łojasiewicz (PL), e.g., see [18] for more details. Therefore, any result based on $\mu$-PL assumption also applies assuming $\mu$-strong convexity. It is noteworthy that while many popular convex optimization problems such as logistic regression and least-squares are often not strongly convex, but satisfy $\mu$-PL condition [18].*

## 4   Convergence Analysis

In this section, we present the convergence analysis of the LUPA-SGD($\tau$) algorithm. All the proofs are deferred to the appendix. We define an auxiliary variable $\bar{\mathbf{x}}^{(t)} = \frac{1}{p}\sum_{j=1}^{p}\mathbf{x}_j^{(t)}$, which is the average model across $p$ different machines at iteration $t$. Using the definition of $\bar{\mathbf{x}}^{(t)}$, the update rule in Algorithm 1, can be written as:

$$\bar{\mathbf{x}}^{(t+1)} = \bar{\mathbf{x}}^{(t)} - \eta\Big[\frac{1}{p}\sum_{j=1}^{p}\tilde{\mathbf{g}}_j^{(t)}\Big], \tag{6}$$

which is equivalent to

$$\bar{\mathbf{x}}^{(t+1)} = \bar{\mathbf{x}}^{(t)} - \eta\nabla F(\bar{\mathbf{x}}^{(t)}) + \eta\Big[\nabla F(\bar{\mathbf{x}}^{(t)}) - \frac{1}{p}\sum_{j=1}^{p}\tilde{\mathbf{g}}_j^{(t)}\Big],$$

thus establishing a connection between our algorithm and the perturbed SGD with deviation $\left(\nabla F(\bar{\mathbf{x}}^{(t)}) - \frac{1}{p}\sum_{j=1}^{p}\tilde{\mathbf{g}}_j^{(t)}\right)$. We show that by i.i.d. assumption and averaging with properly chosen number of local updates, we can reduce the variance of unbiased gradients to obtain the desired convergence rates with linear speed up. The convergence rate of LUPA-SGD($\tau$) algorithm as stated below:

**Theorem 1.** *For LUPA-SGD($\tau$) with $\tau$ local updates, under Assumptions 1 - 3, if we choose the learning rate as $\eta_t = \frac{4}{\mu(t+a)}$ where $a = \alpha\tau + 4$ with $\alpha$ being a constant satisfying $\alpha\exp\left(-\frac{2}{\alpha}\right) < \kappa\sqrt{192(\frac{p+1}{p})}$, and initialize all local model parameters at the same point $\bar{\mathbf{x}}^{(0)}$, for $\tau$ sufficiently large to ensure that[1] that $4(a-3)^{\tau-1}L(C_1 + p) \leq \frac{64L^2(p+1)}{\mu p}(\tau-1)\tau(a+1)^{\tau-2}$, $\frac{32L^2}{\mu}C_1(\tau-1)(a+1)^{\tau-2} \leq \frac{64L^2}{\mu}(\tau-1)\tau(a+1)^{\tau-2}$ and*

$$\tau \geq \frac{\left((\frac{p+1}{p})192\kappa^2 e^{\frac{4}{\alpha}} + 6\alpha\right) + \sqrt{\left((\frac{p+1}{p})192\kappa^2 e^{\frac{4}{\alpha}} + 6\alpha\right)^2 + 20\left((\frac{p+1}{p})192\kappa^2 e^{\frac{4}{\alpha}} - \alpha^2\right)}}{2\left((\frac{p+1}{p})192\kappa^2 e^{\frac{4}{\alpha}} - \alpha^2\right)} \tag{7}$$

*after $T$ iterations we have:*

$$\mathbb{E}\left[F(\bar{\mathbf{x}}^{(T)}) - F^*\right] \leq \frac{\mu B p a^3 \mathbb{E}\left[F(\bar{\mathbf{x}}^{(0)}) - F^*\right] + 4\kappa\sigma^2 T(T + 2a) + 256\kappa^2\sigma^2 T(\tau - 1)}{\mu B p (T + a)^3}, \quad (8)$$

*where $F^*$ is the global minimum and $\kappa = L/\mu$ is the condition number.*

An immediate result of above theorem is the following:

**Corollary 1.** *In Theorem 1 choosing $\tau = O\left(\frac{T^{\frac{2}{3}}}{p^{\frac{1}{3}} B^{\frac{1}{3}}}\right)$ leads to the following error bound:*

$$\mathbb{E}\left[F(\bar{\mathbf{x}}^{(T)}) - F^*\right] \leq O\left(\frac{Bp(\alpha\tau + 4)^3 + T^2}{Bp(T + a)^3}\right) = O\left(\frac{1}{pBT}\right),$$

Therefore, for large number of iterations $T$ the convergence rate becomes $O\left(\frac{1}{pBT}\right)$, thus achieving a linear speed up with respect to the mini-batch size $B$ and the number of machines $p$. A direct implication of Theorem 1 is that by proper choice of $\tau$, i.e., $O\left(T^{\frac{2}{3}}/p^{\frac{1}{3}}\right)$, and periodically averaging the local models it is possible to reduce the variance of stochastic gradients as discussed before. Furthermore, as $\mu$-strong convexity implies $\mu$-PL condition [18], Theorem 1 holds for $\mu$-strongly convex cost functions as well.

## 4.1 Comparison with existing algorithms

Noting that the number of communication rounds is $T/\tau$, for general non-convex optimization, [38] improves the number of communication rounds in [43] from $O(p^{\frac{3}{4}} T^{\frac{3}{4}})$ to $O(p^{\frac{3}{2}} T^{\frac{1}{2}})$. In [33], by exploiting bounded variance and bounded gradient assumptions, it has been shown that for strongly convex functions with $\tau = O(\sqrt{T/p})$, or equivalently $T/\tau = O(\sqrt{pT})$ communication rounds, linear speed up can be achieved. In comparison to [33], we show that using the weaker Assumption 3, for non-convex cost functions under PL condition with $\tau = O\left(T^{\frac{2}{3}}/p^{\frac{1}{3}}\right)$ or equivalently $T/\tau = O\left((pT)^{\frac{1}{3}}\right)$ communication rounds, linear speed up can be achieved. All these results are summarized in Table 1.

The detailed proof of Theorem 1 will be provided in appendix, but here we discuss how a tighter convergence rate compared to [33] is obtainable. In particular, the main reason behind improvement of the LUPA-SGD over [33] is due to the difference in Assumption 3 and a novel technique introduced to prove the convergence rate. The convergence rate analysis of [33] is based on the uniformly bounded gradient assumption, $\mathbb{E}\left[\|\tilde{\mathbf{g}}_j\|_2^2\right] \leq G^2$, and bounded variance, $\mathbb{E}\left[\|\tilde{\mathbf{g}}_j - \mathbf{g}_j\|_2^2\right] \leq \frac{\sigma^2}{B}$, which leads to the following bound on the difference between local solutions and their average at $t$th iteration:

$$\frac{1}{p}\sum_{j=1}^{p}\mathbb{E}\left[\|\bar{\mathbf{x}}^{(t)} - \mathbf{x}_j^{(t)}\|_2^2\right] \leq 4\eta_t^2 G^2 \tau^2. \quad (9)$$

In [33] it is shown that weighted averaging over the term (9) results in the term $O\left(\frac{\kappa\tau^2}{\mu T^2}\right)$ in their convergence bound which determines the maximum allowable size of the local updates without hurting optimal convergence rate. However, our analysis based on the assumption $\mathbb{E}_{\xi_j}\left[\|\tilde{\mathbf{g}}_j - \mathbf{g}_j\|^2\right] \leq C_1\|\mathbf{g}_j\|^2 + \frac{\sigma^2}{B}$, implies the following bound (see Lemma 3 in appendix with $t_c \triangleq \lfloor \frac{t}{\tau}\rfloor\tau$):

$$\sum_{j=1}^{p}\mathbb{E}\|\bar{\mathbf{x}}^{(t)} - \mathbf{x}_j^{(t)}\|^2 \leq 2\left(\frac{p+1}{p}\right)[C_1 + \tau]\sum_{k=t_c}^{t-1}\eta_k^2\sum_{j=1}^{p}\|\nabla F(\mathbf{x}_j^{(k)})\|^2 + 2\left(\frac{p+1}{p}\right)\tau\eta_{t_c}^2\frac{\sigma^2}{B}. \quad (10)$$

Note that we obtain (10) using the non-increasing property of $\eta_t$ from Lemma 3 by careful analysis of the effect of first term in (10) and the weighted averaging. In particular, in our analysis we show that the second term in (10) can be reduced to $\frac{256\kappa^2\sigma^2 T(\tau-1)}{\mu p B(T+a)^3}$ in Theorem 1; hence resulting in improved upper bound over the number of local updates.

# 5 Adaptive LUPA-SGD

The convergence results discussed so far are indicated based on a fixed number of local updates, $\tau$. Recently, [45] and [20] have shown empirically that more frequent communication in the beginning leads to improved performance over fixed communication period.

The main idea behind adaptive variant of LUPA-SGD stems from the following observation. Let us consider the convergence error of LUPA-SGD algorithm as stated in (8). A careful investigation of the obtained rate $O\left(\frac{1}{pT}\right)$ reveals that we need to have $a^3 \mathbb{E}\left[F(\bar{\mathbf{x}}^{(0)}) - F^*\right] = O\left(T^2\right)$ for $a = \alpha\tau + 4$ where $\alpha$ being a constant, or equivalently $\tau = O\left(T^{\frac{2}{3}}/p^{\frac{1}{3}}\left[F(\bar{\mathbf{x}}^{(0)}) - F^*\right]^{\frac{1}{3}}\right)$. Therefore, the number of local updates $\tau$ can be chosen proportional to the distance of objective at initial model, $\bar{\mathbf{x}}^{(0)}$, to the objective at optimal solution, $\mathbf{x}^*$. Inspired by this observation, we can think of the $i$th communication period as if machines restarting training at a new initial point $\bar{\mathbf{x}}^{(i\tau_0)}$, where $\tau_0$ is the number of initial local updates, and propose the following strategy to adaptively decide the number of local updates before averaging the models:

$$\tau_i = \lceil\left(\frac{F(\bar{\mathbf{x}}^{(0)})}{F(\bar{\mathbf{x}}^{(i\tau_0)}) - F^*}\right)^{\frac{1}{3}}\rceil\tau_0 \overset{①}{\to} \tau_i = \lceil\left(\frac{F(\bar{\mathbf{x}}^{(0)})}{F(\bar{\mathbf{x}}^{(i\tau_0)})}\right)^{\frac{1}{3}}\rceil\tau_0, \tag{11}$$

where $\sum_{i=1}^{E} \tau_i = T$ and $E$ is the total number of synchronizations, and ① comes from the fact that $F(\mathbf{x}^{(t)}) \geq F^*$ and as a result we can simply drop the unknown global minimum value $F^*$ from the denominator of (11). Note that (11) generates increasing sequence of number of local updates. A variation of this choice to decide on $\tau_i$ is discussed in Section 6. We denote the adaptive algorithm by ADA-LUPA-SGD$(\tau_1, \ldots, \tau_E)$ for an arbitrary (not necessarily increasing) sequence of positive integers. Following theorem analyzes the convergence rate of adaptive algorithm, ADA-LUPA-SGD $(\tau_1, \ldots, \tau_E)$).

**Theorem 2.** *For ADA-LUPA-SGD $(\tau_1, \ldots, \tau_E)$ with local updates, under Assumptions 1 to 3, if we choose the learning rate as $\eta_t = \frac{4}{\mu(t+c)}$ and all local model parameters are initialized at the same point, for $\tau_i$, $1 \leq i \leq E$ sufficiently large to ensure that $4(c-3)^{\tau_i-1}L(C_1 + p) \leq \frac{64L^2(p+1)}{\mu p}(\tau_i - 1)\tau_i(c+1)^{\tau_i-2}$, $\frac{32L^2}{\mu}C_1(\tau_i - 1)(c+1)^{\tau_i-2} \leq \frac{64L^2}{\mu}(\tau_i - 1)\tau_i(c+1)^{\tau_i-2}$, and $\tau_i, i = 1, 2, \ldots, E$ satisfies the condition in (7), then after $T = \sum_{i=1}^{E} \tau_i$ iterations we have:*

$$\mathbb{E}\left[F(\bar{\mathbf{x}}^{(T)}) - F^*\right] \leq \frac{c^3}{(T+c)^3}\mathbb{E}\left[F(\bar{\mathbf{x}}^{(0)}) - F^*\right] + \frac{4\kappa\sigma^2 T(T+2c)}{\mu Bp(T+c)^3} + \frac{256\kappa^2\sigma^2 \sum_{i=1}^{E}(\tau_i - 1)\tau_i}{\mu Bp(T+c)^3}. \tag{12}$$

*where $c = \alpha \max_{1 \leq i \leq E} \tau_i + 4$, $\alpha$ is a constant satisfying $\alpha \exp\left(-\frac{2}{\alpha}\right) < \kappa\sqrt{192(\frac{p+1}{p})}$, $F^*$ is the global minimum and $\kappa = L/\mu$ is the condition number.*

We emphasize that Algorithm 1 with sequence of local updates $\tau_1, \ldots, \tau_E$, preserves linear speed up as long as the following three conditions are satisfied: i) $\sum_{i=1}^{E} \tau_i = T$, ii) $\sum_{i=1}^{E} \tau_i(\tau_i - 1) = O(T^2)$, iii) $\left(\max_{1 \leq i \leq E} \tau_i\right)^3 = O\left(\frac{T^2}{pB}\right)$. Note that exponentially increasing $\tau_i$ that results in a total of $O(\log T)$ communication rounds, does not satisfy these three conditions. Thus our result sheds some theoretical insight of ADA-LUPA algorithm on how big we can choose $\tau_i$- under our setup and convergence techniques while preserving linear speed up - although, we note that impossibility results need to be derived in future work to cement this insight.

Additionally, the result of [37] is based on minimizing convergence error with respect to the wall-clock time using an adaptive synchronization scheme, while our focus is on reducing the number of communication rounds for a fixed number of model updates. Given a model for wall clock time, our analysis can be readily extended to further fine-tune the communication-computation complexity of [37].

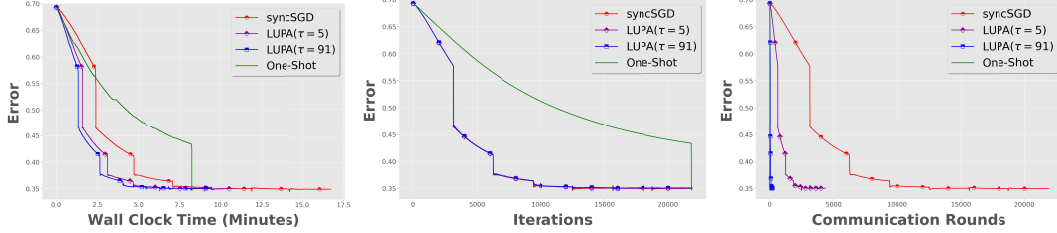

Figure 2: Comparison of the convergence rate of SyncSGD with LUPA-SGD with $\tau = 5$ [33], $\tau = 91$ (ours) and one-shot (with only one communication round).

## 6 Experiments

To validate the proposed algorithm compared to existing work and algorithms, we conduct experiments on Epsilon dataset[2], using logistic regression model, which satisfies PL condition. Epsilon dataset, a popular large scale binary dataset, consists of $400,000$ training samples and $100,000$ test samples with feature dimension of 2000.

**Experiment setting.** We run our experiments on two different settings implemented with different libraries to show its efficacy on different platforms. Most of the experiments will be run on Amazon EC2 cluster with 5 p2.xlarge instances. In this environment we use PyTorch [26] to implement LUPA-SGD as well as the baseline SyncSGD. We also use an internal high performance computing (HPC) cluster equipped with NVIDIA Tesla V100 GPUs. In this environment we use Tensorflow [1] to implement both SyncSGD and LUPA-SGD. The performance on both settings shows the superiority of the algorithm in both time and convergence[3].

**Implementations and setups.** To run our algorithm, as we stated, we will use logistic regression. The learning rate and regularization parameter are $0.01$ and $1 \times 10^{-4}$, respectively, and the mini-batch size is 128 unless otherwise stated. We use mpi4py library from OpenMPI[4] as the MPI library for distributed training.

**Normal training.** The first experiment is to do a normal training on epsilon dataset. As it was stated, epsilon dataset has $400,000$ training samples, and if we want to run the experiment for 7 epochs on 5 machines with mini-batch size of 128 ($T = 21875$), based on Table 1, we can calculate the given value for $\tau$ which for our LUPA-SGD is $T^{\frac{2}{3}}/(pb)^{\frac{1}{3}} \approx 91$. If we follow the $\tau$ in [33] we would have to set $\tau$ as $\sqrt{T/pb} \approx 5$ for this experiment.

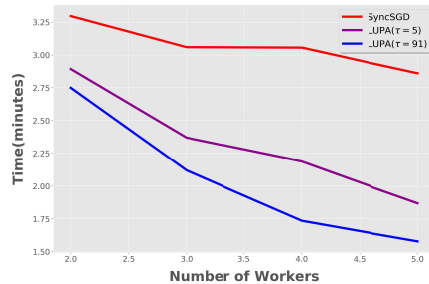

We also include the results for one-shot learning, which is local SGD with only having one round of communication at the end. The results are depicted in Figure 2, shows that LUPA-SGD with higher $\tau$, can indeed, converges to the same level as SyncSGD with faster rate in terms of wall clock time.

**Speedup.** To show that LUPA-SGD with greater number of local updates can still benefits from linear speedup with increasing the number of machines, we run our experiment on different number of machines. Then, we report the time that each of them reaches to a certain error level, say $\epsilon = 0.35$. The results are the average of 5 repeats.

**Adaptive LUPA SGD.** To show how ADA LUPA-SGD works, we run two experiments,

Figure 3: Changing the number of machines and calculate time to reach certain level of error rate ($\epsilon = 0.35$). It indicates that LUPA-SGD with $\tau = 91$ can benefit from linear speedup by increasing the number of machines. The experiment is repeated 5 times and the average is reported.

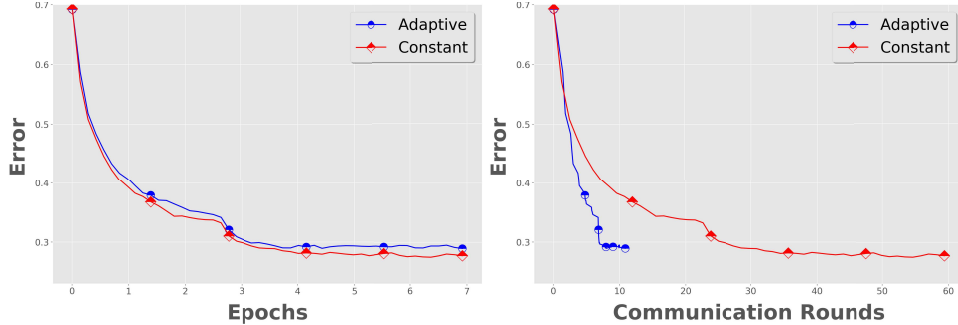

Figure 4: Comparison of the convergence rate of LUPA-SGD with ADA-LUPA-SGD with $\tau = 91$ for LUPA-SGD, and $\tau_0 = 91$ and $\tau_i = (1 + i\alpha)\tau_0$, with $\alpha = 1.09$ for ADA-LUPA-SGD to have 10 rounds of communication. The results show that ADA-LUPA-SGD can reach the same level of error rate as LUPA-SGD, with less number of communication.

first with constant $\tau = 91$ and the other with increasing number of local updates starting with $\tau_0 = 91$ and $\tau_i = (1 + i\alpha)\tau_0$, with $\alpha \geq 0$. We set $\alpha$ in a way to have certain number of communications. This experiment has been run on Tensorflow setting described before.

We note that having access to the function $F(\mathbf{x}^{(t)})$ is only for theoretical analysis purposes and is not necessary in practice as long as the choice of $\tau_i$ satisfies the conditions in the statement of the theorem. In fact as explained in our experiments, we do NOT use the function value oracle and increase $\tau_i$ within each communication period linearly (please see Figure 4) which demonstrates improvement over keeping $\tau_i$ constant.

## 7    Conclusion and Future Work

In this paper, we strengthen the theory of local updates with periodic averaging for distributed non-convex optimization. We improve the previously known bound on the number of local updates, while preserving the linear speed up, and validate our results through experiments. We also presented an adaptive algorithm to decide the number of local updates as algorithm proceeds.

Our work opens few interesting directions as future work. First, it is still unclear if we can preserve linear speed up with larger local updates (e.g., $\tau = O\left(T/\log T\right)$ to require $O\left(\log T\right)$ communications). Recent studies have demonstrated remarkable observations about using large mini-bath sizes from practical standpoint: [41] demonstrated that the maximum allowable mini-batch size is bounded by *gradient diversity* quantity, and [42] showed that using larger mini-batch sizes can lead to superior training error convergence. These observations raise an interesting question that is worthy of investigation. In particular, an interesting direction motivated by our work and the contrasting views of these works would be exploring the maximum allowable $\tau$ for which performance does not decay with fixed bound on the mini-batch size. Finally, obtaining lower bounds on the number of local updates for a fixed mini-bath size to achieve linear speedup is an interesting research question.

## Acknowledgement

This work was partially supported by the NSF CCF 1553248 and NSF CCF 1763657 grants.

## Footnotes

[1]Note that this is a mild condition: if we choose $\tau$ as an increasing function of $T$, e.g., Corollary 1, this condition holds. Note also that $C_1$ can be tuned to be small enough if required via appropriate sampling.

[2]https://www.csie.ntu.edu.tw/~cjlin/libsvmtools/datasets/binary.html

[3]The implementation code is available at https://github.com/mmkamani7/LUPA-SGD.

[4]https://www.open-mpi.org/

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
