[Supplementary Material]

# Supplementary Material
# Local SGD with Periodic Averaging:
# Tighter Analysis and Adaptive Synchronization

**Notation**: In the rest of the appendix, we use the following notation for ease of exposition:

$$\bar{\mathbf{x}}^{(t)} \triangleq \frac{1}{p} \sum_{j=1}^{p} \mathbf{x}_j^{(t)}, \quad \tilde{\mathbf{g}}^{(t)} \triangleq \frac{1}{p} \sum_{j=1}^{p} \tilde{\mathbf{g}}_j^{(t)}, \quad \zeta(t) \triangleq \mathbb{E}[F(\bar{\mathbf{x}}^{(t)}) - F^*], \quad t_c \triangleq \lfloor \frac{t}{\tau} \rfloor \tau \qquad (13)$$

We also indicate inner product between vectors $\mathbf{x}$ and $\mathbf{y}$ with $\langle \mathbf{x}, \mathbf{y} \rangle$.

## A  Proof of Theorem 1

The proof is based on the Lipschitz continuous gradient assumption, which gives:

$$\mathbb{E}\big[F(\bar{\mathbf{x}}^{(t+1)}) - F(\bar{\mathbf{x}}^{(t)})\big] \leq -\eta_t \mathbb{E}\big[\langle \nabla F(\bar{\mathbf{x}}^{(t)}), \tilde{\mathbf{g}}^{(t)} \rangle\big] + \frac{\eta_t^2 L}{2} \mathbb{E}\big[\|\tilde{\mathbf{g}}^{(t)}\|^2\big] \qquad (14)$$

The second term in left hand side of (14) is upper-bounded by the following lemma:

**Lemma 1.** *Under Assumptions 1 and 2, we have the following bound*

$$\mathbb{E}\big[\|\tilde{\mathbf{g}}^{(t)}\|^2\big] \leq \Big(\frac{C_1}{p} + 1\Big) \sum_{j=1}^{p} \|\nabla F(\mathbf{x}_j^{(t)})\|^2 + \frac{\sigma^2}{pB} \qquad (15)$$

The first term in left-hand side of (14) is bounded with following lemma:

**Lemma 2.** *Under Assumptions 3, and according to the Algorithm 1 the expected inner product between stochastic gradient and full batch gradient can be bounded by:*

$$-\eta_t \mathbb{E}\Big[\langle \nabla F(\bar{\mathbf{x}}^{(t)}), \tilde{\mathbf{g}}^{(t)} \rangle\Big] \leq -\frac{\eta_t}{2} \mathbb{E}\big[\|\nabla F(\bar{\mathbf{x}}^{(t)})\|^2\big] - \frac{\eta_t}{2} \frac{1}{p} \sum_{j=1}^{p} \|\nabla F(\mathbf{x}_j^{(t)})\|^2 + \frac{\eta_t L^2}{2p} \mathbb{E} \sum_{j=1}^{p} \|\bar{\mathbf{x}}^{(t)} - \mathbf{x}_j^{(t)}\|^2$$

$$(16)$$

The third term in (16) is bounded as follows:

**Lemma 3.** *Under Assumptions 1 to 2, for $k\tau + 1 \nmid t$ for some $k \geq 1$, we have:*

$$\mathbb{E} \sum_{j=1}^{p} \|\bar{\mathbf{x}}^{(t)} - \mathbf{x}_j^{(t)}\|^2 \leq 2\Big(\frac{p+1}{p}\Big)\Big([C_1 + \tau] \sum_{k=t_c+1}^{t-1} \eta_k^2 \sum_{j=1}^{p} \| \nabla F(\mathbf{x}_j^{(k)})\|^2 + \sum_{k=t_c+1}^{t-1} \frac{\eta_k^2 \sigma^2}{B}\Big) \quad (17)$$

*Note that first this lemma implies that the term $\mathbb{E} \sum_{j=1}^{p} \|\bar{\mathbf{x}}^{(t)} - \mathbf{x}_j^{(t)}\|^2$ only depends on the time $t_c \triangleq \lfloor \frac{t}{\tau} \rfloor \tau$ through $t - 1$. Second, it is noteworthy that since $\bar{\mathbf{x}}^{(t_c+1)} = \mathbf{x}_j^{(t_c+1)}$ for $1 \leq j \leq p$, we have $\mathbb{E} \sum_{j=1}^{p} \|\bar{\mathbf{x}}^{(t_c+1)} - \mathbf{x}_j^{(t_c+1)}\|^2 = 0$.*

Now using the notation $\zeta(t) \triangleq \mathbb{E}[F(\bar{\mathbf{x}}^{(t)}) - F^*]$ and by plugging back all the above lemmas into result (14), we get:

$$\zeta^{(t+1)} \leq (1 - \mu\eta_t)\zeta^{(t)} + \frac{L\eta_t^2 \sigma^2}{2pB} + \frac{\eta_t L^2}{p}\Big(\sum_{k=t_c+1}^{t-1} \eta_k^2 \frac{(p+1)\sigma^2}{pB}\Big) + \frac{\eta_t}{2p}\Big[-1 + L\eta_t(C_1 + p)\Big] \sum_{j=1}^{p} \| \nabla F(\mathbf{x}_j^{(t)})\|^2$$

$$+ \frac{\eta_t L^2}{p}\Big[\Big(C_1(\frac{p+1}{p}) + 2(\tau - 1)\Big) \sum_{k=t_c+1}^{t-1} \sum_{j=1}^{p} \eta_k^2 \| \nabla F(\mathbf{x}_j^{(k)})\|^2\Big]$$

$$\overset{①}{=} \Delta_t \zeta^{(t)} + A_t + D_t \sum_{j=1}^{p} \| \nabla F(\mathbf{x}_j^{(t)})\|^2 + B_t \sum_{k=t_c+1}^{t-1} \eta_k^2 \sum_{j=1}^{p} \| \nabla F(\mathbf{x}_j^{(t)})\|^2, \tag{18}$$

where in ① we use the following from the definitions:

$$\Delta_t \triangleq 1 - \mu\eta_t \tag{19}$$

$$A_t \triangleq \frac{\eta_t L \sigma^2}{pB}\Big[\frac{\eta_t}{2} + \frac{L(p+1)}{p} \sum_{k=t_c+1}^{t-1} \eta_k^2\Big] \tag{20}$$

$$D_t \triangleq \frac{\eta_t}{2p}\Big[ - 1 + L\eta_t(C_1 + p)\Big] \tag{21}$$

$$B_t \triangleq \frac{\eta_t L^2 (p+1)}{p^2}\Big(C_1 + \tau\Big), \tag{22}$$

In the following lemma, we show that with proper choice of learning rate the negative coefficient of the $\|\nabla F(\mathbf{x}_j^{(t)})\|_2^2$ can be dominant at each communication time periodically. Thus, we can remove the terms including $\|\nabla F(\mathbf{x}_j^{(t)})\|_2^2$ from the bound in (18).

Adopting the following notation for $n \leq m$:

$$\mathcal{A}_n^{(m)} = [A_n \quad A_{n+1} \quad \cdots \quad A_{m-1} \quad A_m] \tag{23}$$

$$\mathcal{B}_n^{(m)} = [B_n \quad B_{n+1} \quad \cdots \quad B_{m-1} \quad B_m] \tag{24}$$

$$\Gamma_n^{(m)} = \Pi_{i=n}^{m} \Delta_i \tag{25}$$

$$\mathbf{\Gamma}_n^{(m)} = \Big[\Gamma_n^{(m)} \quad \Gamma_{n+1}^{(m)} \quad \cdots \quad \Gamma_m^{(m)} \quad 1\Big] \tag{26}$$

with convention that $\Gamma_m^{(m)} = \Delta_m$, we have the following lemma:

**Lemma 4.** *We have:*

$$\zeta^{(t+1)} \leq \Gamma_{t_c+1}^{(t)} \zeta^{(t_c+1)} + \Gamma_{t_c+2}^{(t)}\Big[\frac{L\eta_{t_c+1}^2 \sigma^2}{2pB}\Big] + \Big\langle \mathcal{A}_{t_c+1}^{(t)}, \mathbf{\Gamma}_{t_c+3}^{(t)}\Big\rangle$$

$$+ \frac{\eta_t}{2p}\Big[ - 1 + L\eta_t(C_1 + p)\Big]d^{(t)} + \frac{\eta_{t-1}\Delta_t}{2p}\Big[ - 1 + L\eta_{t-1}(C_1 + p) + \frac{2p\eta_{t-1}B_t(\tau - 1)}{\Gamma_t^{(t)}}\Big]d^{(t-1)}$$

$$+ \frac{\Gamma_{t-1}^{(t)}\eta_{t-2}}{2p}\Big[ - 1 + L\eta_{t-2}(C_1 + p) + \frac{2p\eta_{t-2}}{\Gamma_{t-1}^{(t)}}\Big\langle \mathbf{\Gamma}_t^{(t)}, \mathcal{B}_{t-1}^{(t)}\Big\rangle\Big]d^{(t-2)}$$

$$+ \ldots + \frac{\Gamma_{t_c+3}^{(t)}\eta_{t_c+2}}{2p}\Big[ - 1 + L\eta_{t_c+2}(C_1 + p) + \frac{2p\eta_{t_c+2}}{\Gamma_{t_c+3}^{(t)}}\Big\langle \mathbf{\Gamma}_{t_c+4}^{(t)}, \mathcal{B}_{t_c+3}^{(t)}\Big\rangle\Big]d^{(t_c+2)}$$

$$+ \frac{\Gamma_{t_c+2}^{(t)}\eta_{t_c+1}}{2p}\Big[ - 1 + L\eta_{t_c+1}(C_1 + p) + \frac{2p\eta_{t_c+1}}{\Gamma_{t_c+2}^{(t)}}\Big\langle \mathbf{\Gamma}_{t_c+3}^{(t)}, \mathcal{B}_{t_c+2}^{(t)}\Big\rangle\Big]d^{(t_c+1)} \tag{27}$$

**Lemma 5.** *Let $\alpha$ be a positive constant that satisfies $\frac{\alpha}{e^{\frac{2}{\alpha}}} < \kappa\sqrt{192}$ and $a = \alpha\tau + 4$. Suppose that $\tau$ is sufficiently large to ensure that $4(a - 3)^{\tau-1}L(C_1 + p) \leq \frac{64L^2(p+1)}{\mu p}(\tau - 1)\tau(a + 1)^{\tau-2}$, and $\frac{32L^2}{\mu}C_1(\tau - 1)(a+1)^{\tau-2} \leq \frac{64L^2}{\mu}(\tau - 1)\tau(a+1)^{\tau-2}$. If we choose the learning rate as $\eta_t = \frac{4}{\mu(t+a)}$, we have:*

$$\zeta^{(t+1)} \leq \Delta_t \zeta^{(t)} + A_t \tag{28}$$

*for all $1 \leq t \leq T$.*

We conclude the proof of Theorem 1 with the following lemma:

**Lemma 6.** *For the learning rate as given in Lemma 5, iterating over (28) leads to the following bound:*

$$\mathbb{E}[F(\bar{\mathbf{x}}^{(T)}) - F^*] \leq \frac{a^3}{(T+a)^3}\mathbb{E}\big[F(\bar{\mathbf{x}}^{(0)}) - F^*\big] + \frac{4\kappa\sigma^2 T(T+2a)}{\mu p B(T+a)^3} + \frac{256\kappa^2\sigma^2 T(\tau-1)}{\mu p B(T+a)^3} \quad (29)$$

# B  Proof of lemmas

## B.1  Proof of Lemma 1

The proof follows from the Proof of Lemma 6 in [38] by replacing $\sigma^2$ with $\frac{\sigma^2}{B}$.

## B.2  Proof of Lemma 2

Let $\tilde{\mathbf{g}}^{(t)} = \frac{1}{p}\sum_{j=1}^{p}\tilde{\mathbf{g}}_j^{(t)}$. We have:

$$\mathbb{E}\Big[\Big\langle\nabla F(\bar{\mathbf{x}}^{(t)}), \tilde{\mathbf{g}}^{(t)}\Big\rangle\Big] = \mathbb{E}\Big[\Big\langle\nabla F(\bar{\mathbf{x}}^{(t)}), \frac{1}{p}\sum_{j=1}^{p}\tilde{\mathbf{g}}_j\Big\rangle\Big] \quad (30)$$

$$= \frac{1}{p}\sum_{j=1}^{p}\Big[\Big\langle\nabla F(\bar{\mathbf{x}}^{(t)}), \mathbb{E}[\tilde{\mathbf{g}}_j]\Big\rangle\Big] \quad (31)$$

$$\overset{\text{①}}{=} \frac{1}{2}\|\nabla F(\bar{\mathbf{x}}^{(t)})\|^2 + \frac{1}{2p}\sum_{j=1}^{p}\|\nabla F(\mathbf{x}_j^{(t)})\|^2 - \frac{1}{2p}\sum_{j=1}^{p}\|\nabla F(\bar{\mathbf{x}}^{(t)}) - \nabla F(\mathbf{x}_j^{(t)})\|^2$$

$$\overset{\text{②}}{\geq} \mu(F(\bar{\mathbf{x}}^{(t)}) - F^*) + \frac{1}{2p}\sum_{j=1}^{p}\|\nabla F(\mathbf{x}_j^{(t)})\|^2 - \frac{L^2}{2p}\sum_{j=1}^{p}\|\bar{\mathbf{x}}^{(t)} - \mathbf{x}_j^{(t)}\|^2,$$

$$(32)$$

where ① follows from $2\langle\mathbf{a}, \mathbf{b}\rangle = \|\mathbf{a}\|^2 + \|\mathbf{b}\|^2 - \|\mathbf{a} - \mathbf{b}\|^2$ and Assumption 1, and ② comes from Assumption 3.

## B.3  Proof of Lemma 3

Let us set $t_c \triangleq \lfloor\frac{t}{\tau}\rfloor\tau$. Therefore, according to Algorithm 1 we have:

$$\bar{\mathbf{x}}^{(t_c+1)} = \frac{1}{p}\sum_{j=1}^{p}\mathbf{x}_j^{(t_c+1)} \quad (33)$$

for $1 \leq j \leq p$. Then, the update rule of Algorithm 1, can be rewritten as:

$$\mathbf{x}_j^{(t)} = \mathbf{x}_j^{(t-1)} - \eta_{t-1}\tilde{\mathbf{g}}_j^{(t-1)} \overset{\text{①}}{=} \mathbf{x}_j^{(t-2)} - \Big[\eta_{t-2}\tilde{\mathbf{g}}_j^{(t-2)} + \eta_{t-1}\tilde{\mathbf{g}}_j^{(t-1)}\Big] = \bar{\mathbf{x}}^{(t_c+1)} - \Big[\sum_{k=t_c+1}^{t-1}\eta_k\tilde{\mathbf{g}}_j^{(k)}\Big],$$

$$(34)$$

where ① comes from the update rule of our Algorithm. Now, from (34) we compute the average model as follows:

$$\bar{\mathbf{x}}^{(t)} = \bar{\mathbf{x}}^{(t_c+1)} - \Big[\frac{1}{p}\sum_{j=1}^{p}\sum_{k=t_c+1}^{t-1}\eta_k\tilde{\mathbf{g}}_j^{(k)}\Big] \quad (35)$$

First, without loss of generality, suppose $t = t_c + r$ where $r$ denotes the indices of local updates. We note that for $t_c + 1 < t \leq t_c + \tau$, $\mathbb{E}_t\|\bar{\mathbf{x}}^{(t)} - \mathbf{x}_j^{(t)}\|^2$ does not depend on time $t \leq t_c$ for $1 \leq j \leq p$.

We bound the term $\mathbb{E}\|\bar{\mathbf{x}}^{(t)} - \mathbf{x}_l^{(t)}\|^2$ for $t_c + 1 \leq t = t_c + r \leq t_c + \tau$ in three steps: 1) We first relate this quantity to the variance between stochastic gradient and full gradient, 2) We use Assumption 1 on unbiased estimation and i.i.d sampling, 3) We use Assumption 2 to bound the final terms. We proceed to the details each of these three steps.

**Step 1: Relating to variance**

$$\mathbb{E}\|\bar{\mathbf{x}}^{(t_c+r)}-\mathbf{x}_l^{(t_c+r)}\|^2 = \mathbb{E}\|\bar{\mathbf{x}}^{(t_c+1)} - \Big[\sum_{k=t_c+1}^{t-1}\eta_k\tilde{\mathbf{g}}_l^{(k)}\Big] - \bar{\mathbf{x}}^{(t_c+1)} + \Big[\frac{1}{p}\sum_{j=1}^{p}\sum_{k=t_c+1}^{t-1}\eta_k\tilde{\mathbf{g}}_j^{(k)}\Big]\|^2$$

$$\overset{①}{=} \mathbb{E}\|\sum_{k=1}^{r}\eta_{t_c+k}\tilde{\mathbf{g}}_l^{(t_c+k)} - \frac{1}{p}\sum_{j=1}^{p}\sum_{k=1}^{r}\eta_{t_c+k}\tilde{\mathbf{g}}_j^{(t_c+k)}\|^2$$

$$\overset{②}{\leq} 2\Big[\mathbb{E}\|\sum_{k=1}^{r}\eta_{t_c+k}\tilde{\mathbf{g}}_l^{(t_c+k)}\|^2 + \mathbb{E}\|\frac{1}{p}\sum_{j=1}^{p}\sum_{k=1}^{r}\eta_{t_c+k}\tilde{\mathbf{g}}_j^{(t_c+k)}\|^2\Big]$$

$$\overset{③}{=} 2\Big[\mathbb{E}\|\sum_{k=1}^{r}\eta_{t_c+k}\tilde{\mathbf{g}}_l^{(t_c+k)} - \mathbb{E}\big[\sum_{k=1}^{r}\eta_{t_c+k}\tilde{\mathbf{g}}_l^{(t_c+k)}\big]\|^2 + \|\mathbb{E}\big[\sum_{k=1}^{r}\eta_{t_c+k}\tilde{\mathbf{g}}_l^{(t_c+k)}\big]\|^2$$

$$+ \mathbb{E}\|\frac{1}{p}\sum_{j=1}^{p}\sum_{k=1}^{r}\eta_{t_c+k}\tilde{\mathbf{g}}_j^{(t_c+k)} - \mathbb{E}\big[\frac{1}{p}\sum_{j=1}^{p}\sum_{k=1}^{r}\eta_{t_c+k}\tilde{\mathbf{g}}_j^{(t_c+k)}\big]\|^2 + \|\mathbb{E}\big[\frac{1}{p}\sum_{j=1}^{p}\sum_{k=1}^{r}\eta_{t_c+k}\tilde{\mathbf{g}}_j^{(t_c+k)}\big]\|^2$$

$$\overset{④}{=} 2\mathbb{E}\Big(\Big[\|\sum_{k=1}^{r}\eta_{t_c+k}\big[\tilde{\mathbf{g}}_l^{(t_c+k)} - \mathbf{g}_l^{(t_c+k)}\big]\|^2 + \|\sum_{k=1}^{r}\eta_{t_c+k}\mathbf{g}_l^{(t_c+k)}\|^2\Big]$$

$$+ \|\frac{1}{p}\sum_{j=1}^{p}\sum_{k=1}^{r}\eta_{t_c+k}\big[\tilde{\mathbf{g}}_j^{(t_c+k)} - \mathbf{g}_j^{(t_c+k)}\big]\|^2 + \|\frac{1}{p}\sum_{j=1}^{p}\sum_{k=1}^{r}\eta_{t_c+k}\mathbf{g}_j^{(t_c+k)}\|^2\Big),$$

$$(36)$$

where ① holds because $t = t_c + r \leq t_c + \tau$, ② is due to $\|\mathbf{a} - \mathbf{b}\|^2 \leq 2(\|\mathbf{a}\|^2 + \|\mathbf{b}\|^2)$, ③ comes from $\mathbb{E}[\mathbf{X}^2] = \mathbb{E}[[\mathbf{X} - \mathbb{E}[\mathbf{X}]]^2] + \mathbb{E}[\mathbf{X}]^2$, ④ comes from unbiased estimation Assumption 1.

**Step 2: Unbiased estimation and i.i.d. sampling**

$$=2\mathbb{E}\Big(\Big[\sum_{k=1}^{r}\eta_{t_c+k}^2\|\tilde{\mathbf{g}}_l^{(t_c+k)} - \mathbf{g}_l^{(t_c+k)}\|^2$$

$$+ \sum_{w\neq z\vee l\neq v}\Big\langle\eta_w\tilde{\mathbf{g}}_l^{(w)} - \eta_w\mathbf{g}_l^{(w)}, \eta_z\tilde{\mathbf{g}}_v^{(z)} - \eta_z\mathbf{g}_v^{(z)}\Big\rangle + \|\sum_{k=1}^{r}\eta_{t_c+k}\mathbf{g}_l^{(t_c+k)}\|^2\Big]$$

$$+ \frac{1}{p^2}\sum_{l=1}^{p}\sum_{k=1}^{r}\eta_{t_c+k}^2\|\tilde{\mathbf{g}}_l^{(t_c+k)} - \mathbf{g}_l^{(t_c+k)}\|^2$$

$$+ \frac{1}{p^2}\sum_{w\neq z\vee l\neq v}\Big\langle\eta_w\tilde{\mathbf{g}}_l^{(w)} - \eta_w\mathbf{g}_l^{(w)}, \eta_z\tilde{\mathbf{g}}_v^{(z)} - \eta_z\mathbf{g}_v^{(z)}\Big\rangle + \|\frac{1}{p}\sum_{j=1}^{p}\sum_{k=1}^{r}\eta_{t_c+k}\mathbf{g}_j^{(t_c+k)}\|^2\Big)$$

$$\overset{⑤}{=} 2\mathbb{E}\Big(\Big[\sum_{k=1}^{r}\eta_{t_c+k}^2\|\tilde{\mathbf{g}}_l^{(t_c+k)} - \mathbf{g}_l^{(t_c+k)}\|^2 + \|\sum_{k=1}^{r}\eta_{t_c+k}\mathbf{g}_l^{(t_c+k)}\|^2\Big]$$

$$+ \frac{1}{p^2}\sum_{j=1}^{p}\sum_{k=1}^{r}\eta_{t_c+k}^2\|\tilde{\mathbf{g}}_j^{(t_c+k)} - \mathbf{g}_j^{(t_c+k)}\|^2 + \|\frac{1}{p}\sum_{j=1}^{p}\sum_{k=1}^{r}\eta_{t_c+k}\mathbf{g}_j^{(t_c+k)}\|^2\Big)$$

$$\overset{⑥}{\leq} 2\mathbb{E}\Big(\Big[\sum_{k=1}^{r}\eta_{t_c+k}^2\|\tilde{\mathbf{g}}_l^{(t_c+k)} - \mathbf{g}_l^{(t_c+k)}\|^2 + r\sum_{k=1}^{r}\eta_{t_c+k}^2\|\mathbf{g}_l^{(t_c+k)}\|^2\Big]$$

$$+ \frac{1}{p^2}\sum_{j=1}^{p}\sum_{k=1}^{r}\|\tilde{\mathbf{g}}_j^{(t_c+k)} - \mathbf{g}_j^{(t_c+k)}\|^2 + \frac{r}{p^2}\sum_{j=1}^{p}\sum_{k=1}^{r}\eta_{t_c+k}^2\|\mathbf{g}_j^{(t_c+k)}\|^2\Big)$$

$$= 2\Big(\Big[\sum_{k=1}^{r}\eta_{t_c+k}^2\mathbb{E}\|\tilde{\mathbf{g}}_l^{(t_c+k)} - \mathbf{g}_l^{(t_c+k)}\|^2 + r\sum_{k=1}^{r}\eta_{t_c+k}^2\mathbb{E}\|\mathbf{g}_l^{(t_c+k)}\|^2\Big]$$

$$+ \frac{1}{p^2} \sum_{j=1}^{p} \sum_{k=1}^{r} \eta_{t_c+k}^2 \mathbb{E} \|\tilde{\mathbf{g}}_j^{(t_c+k)} - \mathbf{g}_j^{(t_c+k)}\|^2 + \frac{r}{p^2} \sum_{j=1}^{p} \sum_{k=1}^{r} \eta_{t_c+k}^2 \mathbb{E} \|\mathbf{g}_j^{(t_c+k)}\|^2 \Big), \qquad (37)$$

⑤ is due to independent mini-batch sampling as well as unbiased estimation Assumption. ⑥ follow from inequality $\|\sum_{i=1}^{m} \mathbf{a}_i\|^2 \le m \sum_{i=1}^{m} \|\mathbf{a}_i\|^2$.

**Step 3: Using Assumption 2**

Next step is to bound the terms in (37) using Assumption 2 as follow:

$$\mathbb{E} \|\bar{\mathbf{x}}^{(t)} - \mathbf{x}_l^{(t)}\|^2 \le 2 \Big( \Big[ \sum_{k=1}^{r} \eta_{t_c+k}^2 \Big[ C_1 \|\mathbf{g}_l^{(t_c+k)})\|^2 + \frac{\sigma^2}{B} \Big] + r \sum_{k=1}^{r} \eta_{t_c+k}^2 \| \Big[\mathbf{g}_l^{(t_c+k)}\Big]\|^2 \Big]$$

$$+ \frac{1}{p^2} \sum_{j=1}^{p} \sum_{k=1}^{r} \eta_{t_c+k}^2 \Big[ C_1 \|\mathbf{g}_j^{(t_c+k)}\|^2 + \frac{\sigma^2}{B} \Big] + \frac{r}{p^2} \sum_{j=1}^{p} \sum_{k=1}^{r} \eta_{t_c+k}^2 \| \Big[\mathbf{g}_j^{(t_c+k)}\Big]\|^2 \Big)$$

$$= 2 \Big( \Big[ \sum_{k=1}^{r} \eta_{t_c+k}^2 C_1 \|\mathbf{g}_l^{(t_c+k)}\|^2 + \sum_{k=1}^{r} \eta_{t_c+k}^2 \frac{\sigma^2}{B} + r \sum_{k=1}^{r} \eta_{t_c+k}^2 \|\mathbf{g}_l^{(t_c+k)}\|^2 \Big]$$

$$+ \frac{1}{p^2} \sum_{j=1}^{p} \sum_{k=1}^{r} \eta_{t_c+k}^2 C_1 \|\mathbf{g}_j^{(t_c+k)}\|^2 + \sum_{k=1}^{r} \eta_{t_c+k}^2 \frac{\sigma^2}{p^2 B} + \frac{r}{p^2} \sum_{j=1}^{p} \sum_{k=1}^{r} \eta_{t_c+k}^2 \mathbb{E} \|\mathbf{g}_j^{(t_c+k)}\|^2 \Big),$$

$$(38)$$

Now taking summation over worker indices (38), we obtain:

$$\mathbb{E} \sum_{j=1}^{p} \|\bar{\mathbf{x}}^{(t)} - \mathbf{x}_j^{(t)}\|^2 \le 2 \Big( \Big[ \sum_{l=1}^{p} \sum_{k=1}^{r} \eta_{t_c+k}^2 C_1 \|\mathbf{g}_l^{(t_c+k)}\|^2 + \sum_{k=1}^{r} \eta_{t_c+k}^2 \frac{\sigma^2}{B} + r \sum_{l=1}^{p} \sum_{k=1}^{r} \eta_{t_c+k}^2 \|\mathbf{g}_l^{(t_c+k)}\|^2 \Big]$$

$$+ \frac{1}{p} \sum_{j=1}^{p} \sum_{k=1}^{r} \eta_{t_c+k}^2 C_1 \|\mathbf{g}_j^{(t_c+k)}\|^2 + \sum_{k=1}^{r} \eta_{t_c+k}^2 \frac{\sigma^2}{pB} + \frac{r}{p} \sum_{j=1}^{p} \sum_{k=1}^{r} \eta_{t_c+k}^2 \|\mathbf{g}_j^{(t_c+k)}\|^2 \Big)$$

$$= 2 \Big( \Big[ (\frac{p+1}{p}) \sum_{j=1}^{p} \sum_{k=1}^{r} \eta_{t_c+k}^2 C_1 \|\mathbf{g}_j^{(t_c+k)}\|^2 + \sum_{k=1}^{r} \eta_{t_c+k}^2 \frac{(p+1)\sigma^2}{pB}$$

$$+ r(\frac{p+1}{p}) \sum_{j=1}^{p} \sum_{k=1}^{r} \eta_{t_c+k}^2 \|\mathbf{g}_j^{(t_c+k)}\|^2 \Big)$$

$$= 2 \Big( \Big[ (\frac{p+1}{p})(C_1 + r) \Big] \sum_{j=1}^{p} \sum_{k=1}^{r} \eta_{t_c+k}^2 \|\mathbf{g}_j^{(t_c+k)}\|^2 + \sum_{k=1}^{r} \eta_{t_c+k}^2 \frac{(p+1)\sigma^2}{pB} \Big)$$

$$\le 2 \Big( \Big[ (\frac{p+1}{p})(C_1 + \tau) \Big] \Big( \sum_{k=t_c+1}^{t-2} \sum_{j=1}^{p} \eta_k^2 \|\mathbf{g}_j^{(k)}\|^2 + \sum_{j=1}^{p} \eta_{t-1}^2 \|\mathbf{g}_j^{(t-1)}\|^2 \Big) + \sum_{k=t_c+1}^{t-1} \eta_k^2 \frac{(p+1)\sigma^2}{pB} \Big),$$

$$(39)$$

which leads to

$$\mathbb{E} \sum_{j=1}^{p} \|\bar{\mathbf{x}}^{(t)} - \mathbf{x}_j^{(t)}\|^2 \le 2(\frac{p+1}{p}) \Big( [C_1 + \tau] \sum_{k=t_c}^{t-1} \eta_k^2 \sum_{j=1}^{p} \| \nabla F(\mathbf{x}_j^{(k)})\|^2 + \sum_{k=t_c+1}^{t-1} \eta_k^2 \frac{\sigma^2}{B} \Big). \quad (40)$$

### B.4 Proof of Lemma 4

The lemma is simply a recursive application of (18). We write out the details below. We use the short hand notation: $\mathbf{d}^{(t)} \triangleq \sum_{j=1}^{p} \|\nabla \mathbf{F}(\mathbf{x}_j^{(t)})\|^2$.

$$\zeta(t+1) \le \zeta(t) - \mu\eta_t \zeta(t) - \frac{\eta_t}{2p} d^{(t)} + \frac{\eta_t L^2}{2p} \sum_{j=1}^{p} \|\bar{\mathbf{x}}^{(t)} - \mathbf{x}_j^{(t)}\|^2 + \frac{L\eta_t^2}{2p} (\frac{C_1 + p}{p}) d^{(t)} + \frac{L\eta_t^2 \sigma^2}{2pB}$$

$$= (1 - \eta_t \mu)\zeta(t) - \frac{\eta_t}{2p}d^{(t)} + \frac{\eta_t L^2}{2p}\sum_{j=1}^{p}\mathbb{E}\|\bar{\mathbf{x}}^{(t)} - \mathbf{x}_j^{(t)}\|^2 + \frac{L\eta_t^2}{2p}(\frac{C_1 + p}{p})d^{(t)} + \frac{L\eta_t^2\sigma^2}{2pB}$$

$$\overset{\text{①}}{\leq} (1 - \eta_t \mu)\zeta(t) - \frac{\eta_t}{2p}d^{(t)} + \frac{L\eta_t^2}{2}(\frac{C_1 + p}{p})d^{(t)} + \frac{L\eta_t^2\sigma^2}{2pB}$$

$$+ \frac{\eta_t L^2(p+1)}{p^2}\Big[[C_1 + \tau]\sum_{k=t_c+1}^{t-1}\eta_k^2 d^{(k)} + \sum_{k=t_c+1}^{t-1}\eta_k^2\frac{\sigma^2}{B}\Big]$$

$$= (1 - \mu\eta_t)\zeta(t) + \frac{L\eta_t^2\sigma^2}{2pB} + \frac{\eta_t L^2(p+1)\sigma^2}{p^2 B}\sum_{k=t_c+1}^{t-1}\eta_k^2 + \frac{\eta_t}{2p}\Big[-1 + L\eta_t(C_1 + p)\Big]d^{(t)}$$

$$+ \frac{\eta_t L^2(p+1)}{p^2}[C_1 + \tau]\sum_{k=t_c+1}^{t-1}\eta_k^2 d^{(k)}, \tag{41}$$

where ① is due to Lemma 3. Using the notation

$$A_t \triangleq \frac{\eta_t L\sigma^2}{pB}\Big[\frac{\eta_t}{2} + \frac{L(p+1)}{p}\sum_{k=t_c+1}^{t-1}\eta_k^2\Big]$$

$$B_t \triangleq \frac{\eta_t L^2(p+1)}{p^2}[C_1 + \tau]. \tag{42}$$

We can rewrite (41) as follows:

$$\zeta^{(t+1)} \leq (1 - \mu\eta_t)\zeta^{(t)} + A_t + \frac{\eta_t}{2p}\Big[-1 + L\eta_t(C_1 + p)\Big]d^{(t)} + B_t\sum_{k=t_c+1}^{t-1}\eta_k^2 d^{(k)} \tag{43}$$

Now, using the vector notation in (23) and iterating (43), we obtain the following:

$$\zeta^{(t+1)} \leq \Gamma_{t_c+1}^{(t)}\zeta^{(t_c+1)} + \Gamma_{t_c+2}^{(t)}\Big[\frac{L\eta_{t_c+1}^2\sigma^2}{2pB}\Big] + \Big\langle\mathcal{A}_{t_c+1}^{(t)}, \mathbf{\Gamma}_{t_c+3}^{(t)}\Big\rangle$$

$$+ \frac{\eta_t}{2p}\Big[-1 + L\eta_t(C_1 + p)\Big]d^{(t)} + \frac{\eta_{t-1}\Delta_t}{2p}\Big[-1 + L\eta_{t-1}(C_1 + p) + \frac{2p\eta_{t-1}B_t(\tau - 1)}{\Gamma_t^{(t)}}\Big]d^{(t-1)}$$

$$+ \frac{\Gamma_{t-1}^{(t)}\eta_{t-2}}{2p}\Big[-1 + L\eta_{t-2}(C_1 + p) + \frac{2p\eta_{t-2}}{\Gamma_{t-1}^{(t)}}\Big\langle\mathbf{\Gamma}_t^{(t)}, \mathcal{B}_{t-1}^{(t)}\Big\rangle\Big]d^{(t-2)}$$

$$+ \ldots + \frac{\Gamma_{t_c+3}^{(t)}\eta_{t_c+2}}{2p}\Big[-1 + L\eta_{t_c+2}(C_1 + p) + \frac{2p\eta_{t_c+2}}{\Gamma_{t_c+3}^{(t)}}\Big\langle\mathbf{\Gamma}_{t_c+4}^{(t)}, \mathcal{B}_{t_c+3}^{(t)}\Big\rangle\Big]d^{(t_c+2)}$$

$$+ \frac{\Gamma_{t_c+2}^{(t)}\eta_{t_c+1}}{2p}\Big[-1 + L\eta_{t_c+1}(C_1 + p) + \frac{2p\eta_{t_c+1}}{\Gamma_{t_c+2}^{(t)}}\Big\langle\mathbf{\Gamma}_{t_c+3}^{(t)}, \mathcal{B}_{t_c+2}^{(t)}\Big\rangle\Big]d^{(t_c+1)} \tag{44}$$

## B.5 Proof of Lemma 5

To show Lemma 5, it suffices to show that for the choice of learning rates stated in the lemma, the coefficients of $\mathbf{d}^k$ in the statement of Lemma 1, i.e., (27), are all non-positive. So, we aim to show that

$$\eta_t \leq \frac{1}{L(C_1 + p)}$$

$$\eta_{t-1} \leq \frac{1}{L(C_1 + p) + \frac{2pB_t(\tau-1)}{\Gamma_t^{(t)}}}$$

$$\eta_{t-i} \leq \frac{1}{L(C_1 + p) + \frac{2p}{\Gamma_{t-i+1}^{(t)}}\Big\langle\mathbf{\Gamma}_{t-i+2}^{(t)}, \mathcal{B}_{t-i+1}^{(t)}\Big\rangle} \tag{45}$$

for $2 \leq i \leq t - t_c - 1$. Note the following:

1) $\eta_{t_1} > \eta_{t_2}$ if $t_1 < t_2$.

2) $\Delta_{t_1} < \Delta_{t_2}$ if $t_1 < t_2$.

3) $B_{t_1} > B_{t_2}$ if $t_1 < t_2$.

Using these properties, we have:

$$\frac{1}{L(C_1 + p) + \frac{2p}{\Gamma_{t_c+2}^{(t)}}\left\langle \mathbf{\Gamma}_{t_c+3}^{(t)}, \mathcal{B}_{t_c+2}^{(t)} \right\rangle}$$

$$= \frac{1}{L(C_1 + p) + \frac{2p}{\Pi_{i=t}^{t_c+2}\Delta_i}\left[\Pi_{i=t}^{t_c+3}\Delta_i B_{t_c+2} + \ldots + \Delta_t B_{t-1} + B_t\right]}$$

$$\geq \frac{1}{L(C_1 + p) + \frac{2p}{\Pi_{i=t}^{t_c+2}\Delta_i}\left[\Pi_{i=t}^{t_c+3}\Delta_i B_1 + \ldots + \Delta_t B_1 + B_1\right]}$$

$$\overset{⑥}{\geq} \frac{1}{L(C_1 + p) + \frac{2p}{\Delta_1^{\tau-1}}B_1[\tau - 1]}$$

⑥ follows from $\Delta_i \leq 1, i = 1, 2, \ldots, T$.

Since $\eta_t$ is decreasing with $t$, it suffices to show that $\eta_1 \geq \frac{1}{L(C_1+p)+\frac{2p}{\Delta_1^{\tau-1}}B_1[\tau-1]}$. We show that

for the $a = \alpha\tau + 4$ where $\alpha\exp\left(-\frac{1}{\alpha}\right) < \kappa\sqrt{192\left(\frac{p+1}{p}\right)}$ this condition holds. At a high level, note that $\Delta_1^{\tau-1} = (1 - \frac{4}{1+\alpha\tau+4})^{\tau-1}$ is upper bounded by a $e^{4/\alpha}$, that is, as $\tau$ increases, this expression viewed as a function of $\tau$ has a finite limit. Given that $B_1$ is the ratio of two affine terms in $\tau$, we are guaranteed that for a sufficiently small $\alpha$ and for a sufficiently large $\tau$, and performing some elementary manipulations, we can ensure that $\eta_1 = \frac{1}{5+\alpha\tau}$ will be larger than $\frac{1}{L(C_1+p)+\frac{2p}{\Delta_1^{\tau-1}}B_1[\tau-1]} = \frac{1}{\Theta(e^{4/\alpha}\tau)}$. We write out the details below: We aim to show that

$$\eta_1 = \frac{4}{\mu(1 + a)}$$

$$\leq \frac{1}{L(C_1 + p) + \frac{2p}{\Delta_1^{\tau-1}}B_1[\tau - 1]}$$

$$= \frac{\Delta_1^{\tau-1}}{\Delta_1^{\tau-1}L(C_1 + p) + 2pB_1[\tau - 1]}$$

$$= \frac{\left(\frac{1+a-4}{a+1}\right)^{\tau-1}}{\left(\frac{1+a-4}{a+1}\right)^{\tau-1}L(C_1 + p) + 2pB_1[\tau - 1]}$$

$$= \frac{\left(\frac{1+a-4}{a+1}\right)^{\tau-1}}{\left(\frac{1+a-4}{a+1}\right)^{\tau-1}L(C_1 + p) + 2p\left(\frac{4L^2\left(\frac{p+1}{p}\right)(C_1+\tau)}{\mu p(a+1)}\right)(\tau - 1)}$$

$$= \frac{(a - 3)^{\tau-1}}{(a - 3)^{\tau-1}L(C_1 + p) + \left(\frac{p+1}{p}\right)\frac{8L^2}{\mu}\left(C_1(\tau - 1) + (\tau - 1)\tau\right)(a + 1)^{\tau-2}}, \quad (46)$$

Simplifying further, we aim to show that

$$4(a - 3)^{\tau-1}L(C_1 + p) + \frac{32L^2}{\mu}\left(\frac{p+1}{p}\right)\left(C_1(\tau - 1) + \tau(\tau - 1)\right)(a + 1)^{\tau-2}$$

$$\overset{①}{\leq} \frac{192L^2}{\mu^2}\left(\frac{p+1}{p}\right)(\tau - 1)\tau(a + 1)^{\tau-2}$$

$$\leq \mu\left[(1 + a)(a - 3)\right](a - 3)^{\tau-2}, \quad (47)$$

where ① follows from the fact that $(a-3)^{\tau-1}L(C_1+p) \leq \frac{16L^2}{\mu}\tau(\tau-1)(a+1)^{\tau-2}$ and $\frac{32L^2}{\mu}C_1(\frac{p+1}{p})(\tau-1)(a+1)^{\tau-2} \leq (\frac{p+1}{p})\frac{64L^2}{\mu}(\tau-1)^2(a+1)^{\tau-2}$, and the last inequality above has to be shown for sufficiently large $\tau$.

Letting $a = \alpha\tau + 4$ leads to the following condition:

$$\frac{\alpha^2\tau^2 + 6\alpha\tau + 5}{192(\frac{p+1}{p})\frac{L^2}{\mu^2}\tau(\tau-1)} \leq (\frac{a+1}{a-3})^{\tau-2}$$

$$= (1 + \frac{4}{a-3})^{\tau-2}$$

$$= (1 + \frac{4}{\alpha\tau+4-3})^{\tau-2}$$

$$\overset{①}{\leq} e^{\frac{4}{\alpha}}, \tag{48}$$

where ① follows from the property that $\frac{\tau-2}{\alpha\tau+1}$ is non-decreasing with respect to $\tau$. From (48) we get our condition over $\alpha$ as follows:

$$\left((\frac{p+1}{p})192\kappa^2 e^{\frac{4}{\alpha}} - \alpha^2\right)\tau^2 - \left((\frac{p+1}{p})192\kappa^2 e^{\frac{4}{\alpha}} + 6\alpha\right)\tau - 5 \geq 0 \tag{49}$$

. Note that the above is satisfied so long as $\frac{\alpha}{e^{\frac{2}{\alpha}}} \leq \kappa\sqrt{192(\frac{p+1}{p})}$ and

$$\tau \geq \frac{\left((\frac{p+1}{p})192\kappa^2 e^{\frac{4}{\alpha}} + 6\alpha\right) + \sqrt{\left((\frac{p+1}{p})192\kappa^2 e^{\frac{4}{\alpha}} + 6\alpha\right)^2 + 20\left((\frac{p+1}{p})192\kappa^2 e^{\frac{4}{\alpha}} - \alpha^2\right)}}{2\left((\frac{p+1}{p})192\kappa^2 e^{\frac{4}{\alpha}} - \alpha^2\right)}. \tag{50}$$

**Remark 2.** *Note that the left hand side of (46) is independent of the time and is smaller than any condition over $\eta_t$ derived to cancel out the effect of $\|\mathbf{g}\|_2^2$ periodically and satisfying it for every $\eta_t$ is a sufficient condition to have this property.*

Note that due to the choice of $\eta_t$, it can cancel out the effect of $B_t$ and we can rewrite the (43) as follows:

$$\mathbb{E}[F(\bar{\mathbf{x}}^{(t+1)}) - F^*] \leq \Delta_t \mathbb{E}[F(\bar{\mathbf{x}}^{(t)}) - F^*] + A_t \tag{51}$$

### B.6    Proof of Lemma 6

From Lemma 5, we have:

$$\zeta(t+1) \leq \Delta_t\zeta(t) + A_t \tag{52}$$

Define $z_t \triangleq (t+a)^2$ similar to [33], we have

$$\Delta_t\frac{z_t}{\eta_t} = (1-\mu\eta_t)\mu\frac{(t+a)^3}{4} = \frac{\mu(a+t-4)(a+t)^2}{4} \leq \mu\frac{(a+t-1)^3}{4} = \frac{z_{t-1}}{\eta_{t-1}} \tag{53}$$

Now by multiplying both sides of (54) with $\frac{z_t}{\eta_t}$ we have:

$$\frac{z_t}{\eta_t}\zeta(t+1) \leq \zeta(t)\Delta_t\frac{z_t}{\eta_t} + \frac{z_t}{\eta_t}A_t$$

$$\overset{①}{\leq} \zeta(t)\frac{z_{t-1}}{\eta_{t-1}} + \frac{z_t}{\eta_t}A_t, \tag{54}$$

where ① follows from (53). Next iterating over (54) leads to the following bound:

$$\zeta(T)\frac{z_{T-1}}{\eta_{T-1}} \leq (1-\mu\eta_0)\frac{z_0}{\eta_0}\zeta(0) + \sum_{k=0}^{T-1}\frac{z_k}{\eta_k}A_k$$

Final step in proof is to bound $\sum_{k=0}^{T-1}\frac{z_k}{\eta_k}A_k$ as follows:

$$
\begin{aligned}
\sum_{k=0}^{T-1}\frac{z_k}{\eta_k}A_k &= \frac{\mu}{4}\sum_{k=0}^{T-1}(k+a)^3\Big(\frac{L\eta_k^2\sigma^2}{2pB}+\frac{\eta_kL^2}{p}\Big(\sum_{k=t_c+1}^{k-1}\eta_k^2\frac{(p+1)\sigma^2}{pB}\Big)\Big)\\
&\overset{\text{\textcircled{1}}}{\leq}\frac{\mu}{4}\sum_{k=0}^{T-1}(k+a)^3\Big(\frac{L\eta_k^2\sigma^2}{2pB}+\frac{\eta_kL^2}{p}\eta_{\left(\lfloor\frac{k}{\tau}\rfloor\tau\right)}^2(\tau-1)\frac{\sigma^2}{B}\Big(\frac{p+1}{p}\Big)\Big)\\
&=\frac{L\sigma^2\mu}{8pB}\sum_{k=0}^{T-1}(k+a)^3\eta_k^2+\frac{L^2\frac{\sigma^2}{b}(p+1)(\tau-1)\mu}{4p^2}\sum_{k=0}^{T-1}(k+a)^3\eta_k\eta_{\left(\lfloor\frac{k}{\tau}\rfloor\tau\right)}^2,
\end{aligned}
$$
(56)

$\text{\textcircled{1}}$ is due to fact that $\eta_t$ is non-increasing.

Next we bound two terms in (56) as follows:

$$
\begin{aligned}
\sum_{k=0}^{T-1}(k+a)^3\eta_k^2 &= \sum_{k=0}^{T-1}(k+a)^3\frac{16}{\mu^2(k+a)^2}\\
&=\frac{16}{\mu^2}\sum_{k=0}^{T-1}(k+a)\\
&=\frac{16}{\mu^2}\Big(\frac{T(T-1)}{2}+aT\Big)\\
&\leq\frac{8T(T+2a)}{\mu^2},
\end{aligned}
$$
(57)

and similarly we have:

$$
\begin{aligned}
\sum_{k=0}^{T-1}(k+a)^3\eta_k\eta_{\left(\lceil\frac{k}{\tau}\rceil\tau\right)}^2 &= \frac{64}{\mu^3}\sum_{k=0}^{T-1}(k+a)^3\frac{1}{k+a}\Big(\frac{1}{\lfloor\frac{k}{\tau}\rfloor\tau+a}\Big)^2\\
&\overset{\text{\textcircled{1}}}{\leq}\frac{64}{\mu^3}\sum_{k=0}^{T-1}\Big(\frac{k+a}{\lfloor k+a\rfloor}\Big)^2\\
&\overset{\text{\textcircled{2}}}{\leq}\frac{256}{\mu^3}T,
\end{aligned}
$$
(58)

where $\text{\textcircled{1}}$ follows from $\lfloor\frac{k}{\tau}\rfloor\tau+a\geq\lfloor k+a\rfloor$ and $\text{\textcircled{2}}$ comes from the fact that $\frac{n}{\lfloor n\rfloor}\leq 2$ for any integer $n>0$.

Based on these inequalities we get:

$$
\begin{aligned}
\sum_{k=0}^{T-1}\frac{z_k}{\eta_k}A_{k-1}(k) &\leq \frac{L\sigma^2\mu}{8pB}\Big(\frac{8T(T+2a)}{\mu^2}\Big)+\frac{L^2\frac{\sigma^2}{b}(p+1)(\tau-1)\mu}{4p^2}\Big(\frac{256}{\mu^3}T\Big)\\
&=\frac{L\sigma^2T(T+2a)}{pB\mu}+\frac{64L^2\sigma^2T(\tau-1)}{pB\mu^2}\\
&=\frac{\kappa\sigma^2T(T+2a)}{pB}+\frac{64\kappa^2\sigma^2T(\tau-1)}{pB},
\end{aligned}
$$
(59)

Then, the upper bound becomes as follows:

$$
\zeta(T)\frac{z_{T-1}}{\eta_{T-1}}=\mathbb{E}\big[F(\bar{\mathbf{x}}^{(t)})-F^*\big]\frac{\mu(T+a)^3}{4}
$$

$$\leq (1 - \mu\eta_0)\frac{z_{T-1}}{\eta_{T-1}}\zeta(0) + \sum_{k=0}^{T-1}\frac{z_k}{\eta_k}A_k$$

$$\leq (1 - \mu\eta_0)\frac{z_0}{\eta_0}\zeta(0) + \frac{\kappa\frac{\sigma^2}{b}T(T+2a)}{pB} + \frac{64\kappa^2\sigma^2T(\tau-1)}{pB}$$

$$\leq \frac{\mu a^3}{4}\mathbb{E}\big[F(\bar{\mathbf{x}}^{(0)}) - F^*\big] + \frac{\kappa\sigma^2T(T+2a)}{pB} + \frac{64\kappa^2\sigma^2T(\tau-1)}{pB}, \qquad (60)$$

Finally, from (60) we conclude:

$$\mathbb{E}\big[F(\bar{\mathbf{x}}^{(t)}) - F^*\big] \leq \frac{a^3}{(T+a)^3}\mathbb{E}\big[F(\bar{\mathbf{x}}^{(0)}) - F^*\big] + \frac{4\kappa\sigma^2T(T+2a)}{\mu pB(T+a)^3} + \frac{256\kappa^2\sigma^2T(\tau-1)}{\mu pB(T+a)^3}, \quad (61)$$

## C  Proof of Theorem 2

Theorem 2 can be seen as an extension of Theorem 1, and for the purpose of the proof and letting $t_c = \lfloor\frac{t}{\tau_i}\rfloor\tau_i$ where $T = \sum_{i=1}^{E}\tau_i$, we only need following Lemmas:

**Lemma 7.** *Under Assumptions 1 to 3 we have:*

$$\mathbb{E}\sum_{j=1}^{p}\|\bar{\mathbf{x}}^{(t)} - \mathbf{x}_j^{(t)}\|^2 \leq 2(\frac{p+1}{p})\Big([C_1 + \tau_i]\sum_{k=t_c}^{t-1}\eta_k^2\sum_{j=1}^{p}\|\nabla F(\mathbf{x}_j^{(k)})\|^2 + \sum_{k=t_c+1}^{t-1}\eta_k^2\frac{\sigma^2}{B}\Big), \quad (62)$$

**Lemma 8.** *Under assumptions 1 to 3, if we choose the learning rate as $\eta_t = \frac{4}{\mu(t+c)}$ inequality (18) reduces to*

$$\mathbb{E}[F(\bar{\mathbf{x}}^{(t+1)})] - F^* \leq \Delta_t\mathbb{E}[F(\bar{\mathbf{x}}^{(t)}) - F^*] + A_t, \qquad (63)$$

*for all iterations and $c = \alpha\max_i\tau_i + 4$ and $\frac{\alpha}{e^{\frac{1}{\alpha}}} < L\sqrt{\frac{192}{\mu}}$.*

Finally, for the rest of the proof we only need to reconsider the last term as follows:

$$\sum_{k=0}^{T-1}(k+c)^3\eta_k\eta_{(t_c)}^2(\tau_{t_c}-1) = \sum_{i=1}^{E}(\tau_i-1)\sum_{k=1}^{\tau_i}(k+c)^3\frac{4}{\mu(k+c)}\Big(\frac{4}{\mu(\lfloor\frac{k}{\tau_i}\tau_i\rfloor+c)}\Big)^2$$

$$\leq \frac{64}{\mu^3}\sum_{i=1}^{E}(\tau_i-1)\sum_{k=1}^{\tau_i}\Big(\frac{k+c}{\lfloor k+c\rfloor}\Big)^2$$

$$\leq \frac{256}{\mu^3}\sum_{i=1}^{E}(\tau_i-1)\tau_i, \qquad (64)$$

The rest of the proof is similar to the proof of Theorem 1.