[Reviews · NeurIPS 2019]

Reviewer 1



This paper studied the local updates with periodic averaging to solve distributed non-convex optimization problems, and provided an improved bound over the number of required local updates. The theories are validated through experimental results. Overall, I feel the paper contains new results and insights, and is well-organized. I have comments in the following three aspects. Originality: 1. The upper bound of the required local updates is new, which improves the existing one by a constant order. And it is good that the paper also explained why this improvement can be achieved. 2. The adaptive communication rule also seems to be new, but it needs further explanation. Specifically, only the convergence is proved under the adaptive communication rule, which is not that surprising. Is there any theoretical improvement? 3. How does the adaptive communication rule relate to/differ from the adaptive communications rules in the following work? Chen, Tianyi, Georgios Giannakis, Tao Sun, and Wotao Yin. "LAG: Lazily aggregated gradient for communication-efficient distributed learning." In Advances in Neural Information Processing Systems, pp. 5050-5060. 2018. Wang, Jianyu, and Gauri Joshi. Adaptive communication strategies to achieve the best error-runtime trade-off in local-update SGD." arXiv preprint:1810.08313, Oct. 2018. 4. It is mentioned that under the current analysis, the logarithmic number of communications may be not possible, it is encouraged to comment on whether the bound is tight or match its lower bound. ==== Rebuttal ==== I have carefully read the rebuttal. Thanks.

Reviewer 2



The paper tightens the analysis of local SGD that periodically averages the models at different nodes. It improves the bound on the communication rounds suffice to achieve linear speedups and relaxes some of the assumptions used in previous works. The authors also develop an adaptive scheme to choose the communication rounds based on the intuition from their theoretical results. They also provided empirical results on logistic regression problem with epsilon dataset to support the theoretical results. Comments: The paper is well written and east to follow. Although, I haven't checked all the proofs in detail, I believe one can definitely relax the assumptions such as bounded gradients and variances in the analysis which is standard in analyzing centralized sgd and distributed gradient descent methods (smooth problems case). While the authors consider non-convex problems, they restrict the analysis to the ones that satisfy P-L condition which is very similar to that of strong convexity (considered in previous works) and the analysis for both these cases would be exactly same if same assumptions are used. The main theoretical result on communication rounds is better than the previous result in terms of the dependence on total iterations (T) and the number of workers (p). While the motivation for the adaptive scheme is convincing, the practicality of such a scheme after approximations is questionable. For example, given that this is a stochastic problem, do we have access to evaluate functions F in eq 10. If one follows a different scheme, then what are the criteria on which one can select $tau$'s? Like on line 245-246 on page 6? If so, is there an optimal strategy? The empirical results are promising; however, I believe a more thorough numerical investigation would help the theoretical claims. For example, while the communication plots in figure 2 are understandable, it is surprising that syncSGD and LUPA have similar performance in terms of iterations. The theoretical rates might be similar for both of them, but the constants are different. This raises the question of whether or not the learning rate is tuned separately for each of them. If not, then it would not be a fair comparison. Also, more experiments on the same problem, and other problems would strengthen the paper. On another note, there is a lot of work on efficient communication strategies in distributed network settings. One way to look at this problem is as a particular case of fully connected networks with delayed communications. After Authors' Response: I have carefully read the response. Thanks

Reviewer 3



This work presents a much stronger result than the state-of-the-art analysis [14], and should be interesting to stochastic optimization community. The idea is simple and easy to follow. I have only one minor concern: It is desirable to discuss or empirically verify whether the established bound is optimal. Typos: line 423 & 425 ft-handsideright-hand side’

[Author Response · NeurIPS 2019]

Many thanks to the reviewers for their deep, thoughtful reviews and constructive suggestions.

**R1. ADA-LUPA clarification & Advantages:** Thanks for the suggestion. We will elaborate more on the adaptive algorithm. We note that despite very recent observations on empirical superiority of adaptive synchronization (e.g., Figure 4 in our experiments, and [12], and [40] that demonstrate adaptivity can reduce the number of communication in minimizing wall-clock time or leads to a better generalization error), it lacks theoretical understanding. Indeed, Theorem 2 in our paper is the first non-asymptotic analysis of convergence of adaptive local SGD which matches the improved non-additive counterpart. Surely, it would be interesting to see if our bound can be tightened.

**R1. Ada-LUPA vs. existing works:** Thanks for pointing out these references. Here we briefly highlight a few key differences and will add a detailed comparison in the subsequent version of our paper: 1) The result of [Wang & Joshi] is based on minimizing convergence error with respect to the wall-clock time using an adaptive synchronization schema, while our focus is on reducing the number of communications in terms of number of iterations. Obviously, our analysis can be extended to take into account the wall-clock time into consideration to further improve the communication-computation complexity of [Wang & Joshi], and 2) The convergence analysis of their algorithm is asymptotic in essence, while ours is non-asymptotic. The LAG algorithm proposed in [Chen et al], aims at solving the distributed optimization as low communication **overhead** as possible in an adaptive manner (skipping over some local gradients and using outdated gradients instead), while ADA-LUPA reduces the **number** of communication rounds by reducing communication frequency. It is noticeable that the adaptive schema used in [Wang & Joshi] is different from ours as it starts with infrequent averaging to improve convergence speed, and then increases the communication frequency in order to achieve a low error floor. Our schema is consistent with [40] which uses frequent communication at the beginning (warm-up stage) and then infrequent communication with **fixed** number of local updates to reduce the number of communications as we aimed for, but the empirical results are not supported by theoretical analysis.

**R1.** $\log$ **T communication rounds clarification:** At first glance, given that adaptively reducing the communication frequency works well in practice, it might seem that in our adaptive schema by exponentially increasing the number of local updates, one can get linear speedup with log T number of communications. However, the main theoretical insight of ADA-LUPA algorithm is to provide some intuition on how big we can choose $\tau_i$– under our setup and convergence techniques while preserving linear speed up. To see this let $\tau_i = a\tau_{i-1}$ for some $a > 1, i \geq 2$, which means that $O(\log(\frac{Ta}{\tau_1}))$ communication rounds is needed. However, according to Theorem 2, this choice of $\tau_i$ does not allow linear speed up with respect to the number of workers. Quantifying improvement of ADA-LUPA over LUPA from a theoretical standpoint or requiring log T communications with some other tweaks (e.g., using increasing batch sizes) are interesting future directions that are worthy of investigation.

**R2. P-L vs. strong-convexity & proof novelty:** We agree with the reviewer that the convergence proof of *non-local* gradient descent based algorithms with P-L condition is similar or even simpler than the strong-convexity based analysis. However, for local SGD with periodic averaging the proof techniques are more involved. The key challenge is to periodically cancel out the effect of growing $\|\mathbf{g}\|_2^2$ from upper bound (Lemma 4) which requires a different set of novel techniques which distinguishes our analysis from analysis of non-local methods. **We note that even under strong-convexity assumption and removing bounded gradient assumption we obtain an improved communication complexity while preserving the same convergence upper compared to [14] due to our tight analysis as pointed out in the paper.** Regarding the applicability, we remark that while many convex optimization problems (like linear or logistic regression) does not satisfy strong convexity generally, they satisfy P-L condition (e.g. see [15]). Also, the P-L condition allows the generalization of convergence analysis to general non-convex optimization problems similar to [43], which we leave as future work.

**R2. Practicality of ADA-LUPA & optimality:** Many thanks for your meticulous attention. We will make this point clear in the statement of the theorem. We note that having access to the function $F(\mathbf{x}^{(t)})$ is only for theoretical analysis purposes and is not necessary in practice as long as the choice of $\tau_i$ satisfies the conditions in the statement of the theorem. In fact as it is explained in our experiments, we do NOT use the function value oracle and increase $\tau_i$ within each communication period linearly (please see Figure 4) which demonstrates huge improvement. However, we believe that with a more intelligent adjustment we can achieve faster convergence, hence, we will investigate other adjustment schemes and their effects as well. Also, as indicated in [Wang & Joshi] having access to the zero oracles of functions could be possible and we believe it is possible to derive some optimality criteria similar to [Wang & Joshi].

**R2. Numerical investigation & more experiments:** We do not tune the learning rate. The goal of experiments is simply to show the effectiveness of our algorithm compared to other baselines such as syncSGD. Hence, we kept all the hyperparameters the same for different experiments in this figure to have a fair comparison. Although the experiments on this dataset are showing promising results, we will definitely run on several more datasets with different loss functions to better understand the effects of local updates and make our case stronger.

**R3. Optimality:** We note that unlike non-local distributed methods, the communication complexity of local SGD is not well-understood. Although our work makes a step towards tightening the state-of-the-art communication complexity, as our empirical results demonstrated, it could be further improved. That being said, our goal is to achieve a linear speed up with the smallest mini-batch size and the largest possible $\tau$ as the measure of optimality, considering there key parameters p,b, and $\tau$ involved in our convergence error. If we weaken these goals either by using larger mini-batch size as done in [43] or giving up on linear speed up, our convergence analysis can be extended to show convergence with a much smaller number of communication rounds. We will add some empirical experiments regarding the optimality of the bound for the subsequent version and leave the theoretical understanding as an interesting open question.

[Meta-Review · NeurIPS 2019]

The paper has three interesting contributions: Theoretical contributions: The paper improved the upper bound over the number of communication rounds for non-convex optimization problems under Polyak-Lojasiewicz conditions. While the extension from strong convexity to PL is somewhat expected, the paper is clearly written and handles nicely the "relaxing bounded gradients" assumption. Algorithmic contributions: An adaptive scheme for choosing the communication frequency has been developed. One reviewer raised that the adaptive scheme should have been developed with more details, and that could have been an even more significant contribution (new algorithm). Empirical contributions: Experimental results on Amazon EC2 cluster and an internal GPUs cluster is definitely a plus